# REPETITIVE CONTRASTIVE LEARNING ENHANCES MAMBA'S SELECTIVITY IN TIME SERIES PREDICTION

## ABSTRACT

The prediction of long sequences has always been a challenge in time series forecasting tasks. Due to Mamba's sequence selection capability, many Mamba-based models have been proposed, achieving state-of-the-art results in long sequence prediction problems. However, much research has focused on integrating mamba-ssm into specific model structures for better performance, while the core of mamba-ssm, its sequence selection capability, has not been deeply explored. We believe there is significant potential in Mamba's sequence selection capability and propose a Repetitive Contrastive Learning (RCL) method to enhance it. Specifically, we utilize Repeating Sequence Augmentation to expand the sequence while introducing Gaussian noise, thereby enhancing the Mamba block's sequence selection capability through both inter-sequence and intra-sequence contrastive techniques. Then our methods transfer parameters directly from a single pre-trained Mamba block to a variety of Mamba-based models. This approach provides superior initialization for forecasting tasks. Our experiments consistently demonstrate that this technique improves the forecasting performance of many Mamba-based models, without imposing additional memory requirements.

## 1 INTRODUCTION

Time series forecasting (TSF) has become indispensable across a range of critical domains, including financial markets Li et al. (2023), traffic management Cheng et al. (2023), electricity consumption prediction Sun & Zhang (2023), scientific computing Cruz-Camacho et al. (2024), and weather forecasting Zhang et al. (2022a). TSF leverages sequential data, often of varying lengths, from past observations to predict future trends. However, fully reliable predictors remain elusive due to the unknown generative mechanisms underlying time series data. Compounding this challenge are issues such as uneven sampling, missing or duplicate data points, and inherent irregular noise, which complicate the forecasting tasks.

Deep learning has made significant strides in the time series domain, with much of the focus centered on model architecture design, particularly for transformer-based models Wen et al. (2023) . These models now play a pivotal role in forecasting tasks, yet they are often hindered by the quadratic complexity of their attention mechanisms. While time series data share some structural similarities with natural language, transformers typically underperform in this domain compared to traditional backbones like CNNs and MLPs Zeng et al. (2022). Despite the limitations of traditional backbones in capturing long-range dependencies, they are more effective at addressing the sequential and high-noise characteristics of time series data, which contributes to their better performance.

The emergence of the Mamba model Gu & Dao (2024) has attracted researchers from diverse fields, including those focused on multi-modal and multi-dimensional data, thanks to its unique selective state-space mechanism Huang et al. (2024); Li et al. (2024). Mamba's selective mechanism not only resolves the quadratic time complexity of transformer attention mechanisms but also maintains comparable long-distance propagation capabilities. Recent applications of Mamba in TSF, such as TimeMachine Ahamed & Cheng (2024) and Bi-Mamba Liang et al. (2024), have primarily focused on refining Mamba's block architecture. However, these efforts have overlooked the critical challenge of teaching models how to effectively select and prioritize key moments in time series data.

One major reason for this gap is rooted in the training objectives used for TSF. Most existing models focus on straightforward prediction tasks, analogous to causal and masked language modeling in

NLP. Although these methods have yielded undeniable success in natural language processing, they may not be as effective for time series data. NLP tasks often rely on models to grasp contextual knowledge, such as common sense, syntax, and semantics, from the input text. In contrast, time series data lack generalized knowledge, focusing instead on sequential and often sparse patterns. Unlike NLP data, which can be pre-processed to remove noise and ensure consistency, time series data is marked by irregular and noisy sampling points that are often imperceptible to humans and can only be effectively classified by models during training.

Thus, directly applying modules, which are designed for NLP tasks, to time series data without adapting the training goals leads to predictable failures. Recent experiments have shown that relying solely on prediction tasks does not enable Mamba to fully resolve the complexities of time series data Zhang et al. (2024). These results suggest that prediction tasks alone are insufficient for models to gain a deep understanding of the underlying structure and dynamics of time series. In such cases, models often struggle to selectively focus on relevant moments, instead over-integrating all available information in an attempt to capture causal relationships.

Drawing from these insights, we propose that the central task in time series processing should emphasize teaching models effective selection mechanisms. By leveraging Mamba's new selective space models module, we believe it has the potential to surpass other models in handling selection tasks on sparse time series data. To address existing challenges, we introduce a new training paradigm aimed at optimizing Mamba's selection process.

Technically, our approach enhances time series data through Repeating Sequence Augmentation and pre-trains the original models using Repetitive Contrastive Learning. During the augmentation stage, each token in the time series is duplicated and augmented with Gaussian noise. In the learning stage, we implement intra-sequence contrast, which forms positive and negative pairs from corresponding tokens between the augmented sequence and the original sequence, and inter-sequence contrast, which derives positive and negative pairs from within the augmented sequence itself.

We substitute Mamba Block parameters across various Mamba-based models with those obtained from the pre-trained Mamba block and evaluate the resulting performance improvements relative to the original models. Our work also details effective block substitution methods and parameter-freezing strategies.

In summary, our main contributions are as follows:

- We propose a data augmentation method called **Repeating Sequence Augmentation**, which generates extended time series and forms token-level contrastive samples.
- We introduce **Repetitive Contrastive Learning**, which compares original and repeated sequences within a Mamba block to obtain initialization parameters with enhanced sequence selection and robustness.
- We transfer pre-trained parameters to different Mamba-based models, demonstrating a **generalized approach** that consistently improves the predictive capability of these models on various datasets.
- We analyze effective parameter replacement methods, parameter freezing techniques, and the impact on training time and memory overhead.

## 2 PRELIMINARY

### 2.1 MULTIVARIATE TIME SERIES FORECASTING

Multivariate time series forecasting involves predicting future values of multiple interrelated time-dependent variables based on their historical data. Unlike univariate time series forecasting, which focuses on a single variable, multivariate forecasting accounts for interactions and correlations between multiple variables to improve prediction accuracy and insightfulness.

A multivariate time series forecasting problem can be formally represented with an input time series denoted as $\boldsymbol{X} \in \mathbb{R}^{T_{\text{in}} \times F}$, where $T_{\text{in}}$ is the input sequence length (number of time steps) and $F$ represents the number of features or variables at each time step. The prediction target is represented as $\boldsymbol{Y} \in \mathbb{R}^{T_{\text{out}} \times F}$, where $T_{\text{out}}$ denotes the output sequence length for which forecasts are made.

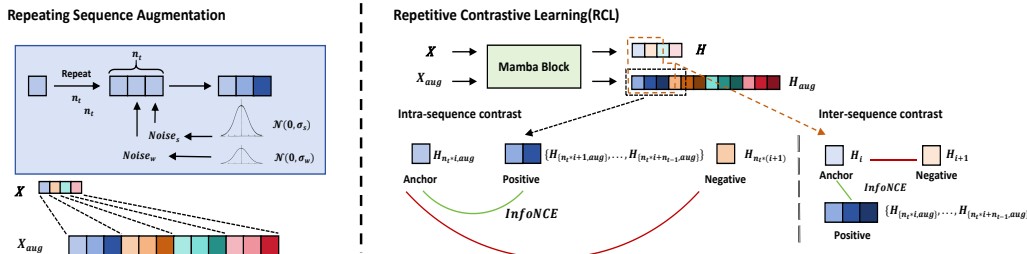

Figure 1: Process of the proposed method. Including Repeating Sequence Augmentation and Repetitive Contrastive Learning (RCL), with RCL consisting of Intra-sequence contrast and Inter-sequence contrast.

## 2.2 TIME SERIES AND INVERTED TIME SERIES

In the latest research, treating the features of a time series as tokens and embedding along the time has proven effective for multivariate time series prediction problems.Liu et al. (2024) Consequently, many models utilize an inverted time series as input. Here, $F$ represents the number of features or variables in the time series. To facilitate distinction between sequences, we define the inverted time series as $\boldsymbol{X}^I \in \mathbb{R}^{F \times T_{\text{in}}}$, which is obtained by transposing the original time series $\boldsymbol{X} \in \mathbb{R}^{T_{\text{in}} \times F}$.

## 3 METHOD

Our proposed repetitive contrastive pre-training method involves three main steps. First, we construct augmented data by repeating time steps and introducing noise, defining positive and negative sample pairs in the process. Next, a Mamba block undergoes pre-training through contrastive learning to enhance its ability to select relevant time series features. Finally, the pre-trained parameters of the Mamba block are transferred to various Mamba-based models for fine-tuning.

## 3.1 REPEATING SEQUENCE AUGMENTATION

One significant reason why Mamba performs exceptionally well in time series prediction tasks is its selective structure. To enhance the selection capability of the Mamba Block, we designed the Repeating Sequence Augmentation. Specifically, as shown in Fig. 1, for each time step in each time series, we sequentially repeat this time step with repetition count $n_t$.

$$\boldsymbol{X}_i \xrightarrow{\text{repeat}} \boldsymbol{X}_{i,1}, ..., \boldsymbol{X}_{i,n_t}$$
$$\boldsymbol{X}_{\text{rep}} = \{\boldsymbol{X}_{1,1}, ..., \boldsymbol{X}_{1,n_t}, ..., \boldsymbol{X}_{i,1}, ..., \boldsymbol{X}_{i,n_t}, ..., \boldsymbol{X}_{s,1}, ..., \boldsymbol{X}_{s,n_t}\}$$

(1)

where $\boldsymbol{X}_i$ is the $i$-th step in time sequence, and $s$ is the length of the sequence. For the time series $\boldsymbol{X} \in \mathbb{R}^{T \times F}$, $s = T$, the corresponding $\boldsymbol{X}_{\text{rep}} \in \mathbb{R}^{(n_t * T) \times F}$. As for inverted time series $\boldsymbol{X}^I \in \mathbb{R}^{F \times T}$, $s = F$, the corresponding $\boldsymbol{X}_{\text{rep}}^I \in \mathbb{R}^{(n_t * F) \times T}$.

Then, we add Gaussian noise of increasing intensity, from weak to strong, to the repeated time steps. In our experiments, we choose $n_t = 3$, each time step $X_i$ is repeated and obtain $\boldsymbol{X}_{i,1}, \boldsymbol{X}_{i,2}, \boldsymbol{X}_{i,2}$. We then sample a strong Gaussian noise and a weak Gaussian noise, and add them to the repeated time steps in increasing order of intensity, from weak to strong.

$$\text{Noise}_\alpha \sim \mathcal{N}(0, \sigma_\alpha^2)$$
$$\text{Noise}_\beta \sim \mathcal{N}(0, \sigma_\beta^2)$$
$$\sigma_\alpha < \sigma_\beta$$
$$\hat{\boldsymbol{X}}_{i,2} = \boldsymbol{X}_{i,2} + \text{Noise}_\alpha \tag{2}$$
$$\hat{\boldsymbol{X}}_{i,3} = \boldsymbol{X}_{i,3} + \text{Noise}_\beta$$
$$\boldsymbol{X}_{\text{aug},i} = \{\boldsymbol{X}_{i,1}, \hat{\boldsymbol{X}}_{i,2}, \hat{\boldsymbol{X}}_{i,3}\}$$
$$\boldsymbol{X}_{\text{aug}} = \boldsymbol{X}_{\text{aug},1} \| \boldsymbol{X}_{\text{aug},2} \| \ldots \| \boldsymbol{X}_{\text{aug},s}$$

where $\text{Noise}_\alpha$ and $\text{Noise}_\beta$ represent weak and strong Gaussian noise, controlled by the variances $\sigma_\alpha$ and $\sigma_\beta$. Since the effect of noise accumulates progressively with the sequential modeling, gradually increasing the noise effectively enlarges the distance between time steps, enhancing the difficulty of the subsequent contrastive learning task.

## 3.2 Mamba Block and IMamba Block

The Mamba block, Gu & Dao (2024), consists of two parts : selection and State Space Model (SSM), as shown in Fig. 2. Firstly, the input $\boldsymbol{X}$ undergoes a one-dimensional convolution (Conv1d) to extract local features, followed by Linear Projection that maps it to matrices $\boldsymbol{B}$, $\boldsymbol{C}$, and $\Delta$.

$$\boldsymbol{X}_c = \sigma(\text{Conv1d}(\boldsymbol{X}))$$
$$\boldsymbol{B} = \text{fc}(\boldsymbol{X}_c), \quad \boldsymbol{C} = \text{fc}(\boldsymbol{X}_c) \tag{3}$$
$$\Delta = \text{softplus}(\text{fc}(\boldsymbol{X}_c) + \boldsymbol{A})$$

where $\sigma$ is SiLU activation function and softplus means the Softplus activation functions, and $\boldsymbol{A}$ is an optimizable matrix. Then, matrices $\boldsymbol{A}$ and $\boldsymbol{B}$ are discretized into $\overline{\boldsymbol{A}}$, $\overline{\boldsymbol{B}}$,

$$\overline{\boldsymbol{A}} = \exp(\Delta \boldsymbol{A})$$
$$\overline{\boldsymbol{B}} = (\exp(\Delta \boldsymbol{A}) - \boldsymbol{I})(\Delta \boldsymbol{A})^{-1}(\Delta \boldsymbol{B}) \tag{4}$$

Finally, Mamba inputs $\overline{\boldsymbol{A}}$, $\overline{\boldsymbol{B}}$, $\boldsymbol{C}$, $\Delta$ and $\boldsymbol{X}$ into the SSM, and uses residual connections.

$$\boldsymbol{H} = \text{SSM}(\overline{\boldsymbol{A}}, \overline{\boldsymbol{B}}, \Delta, \boldsymbol{X}) \cdot \sigma(\text{fc}(\boldsymbol{X})) \tag{5}$$

where fc is fully connected layers, and $\sigma$ is SiLU activation function.

The iMamba block retains the same structural design as the traditional Mamba block; however, the Mamba block processes embedded time vectors along the feature dimension. In contrast, iMamba operates on an inverted time series $\boldsymbol{X}^I$, where each token represents a feature in the time series data. The resulting output is the corresponding sequence $\boldsymbol{H}^I$. This output is then passed through linear layers applied to the embedded time vectors to achieve the desired output length.

## 3.3 Repetitive Contrastive Learning

We input both the original sequence $\boldsymbol{X}$ and its augmented version $\boldsymbol{X}_{\text{aug}}$ into the same Mamba Block, comparing their respective outputs $\boldsymbol{H}$ and $\boldsymbol{H}_{\text{aug}}$ to evaluate the Mamba Block's modeling capabilities across both sequences. As illustrated in Fig.1, Repetitive Contrast Learning (RCL) encompasses two types of comparisons: intra-sequence contrast and inter-sequence contrast. Firstly, we define the output at any time step $i$ with a repetition count $n_t$ of the original sequence $\boldsymbol{X}_i$ as $\boldsymbol{H}_i$, and the output at the subsequent time step as $\boldsymbol{H}_{i+1}$. The outputs of the augmented sequence are represented as $\{\boldsymbol{H}_{\{i \cdot n_t, \text{aug}\}}, \boldsymbol{H}_{\{i \cdot n_t+1, \text{aug}\}}, \ldots, \boldsymbol{H}_{\{i \cdot n_t+n_t-1, \text{aug}\}}\}$, while the output at the next time step is $\{\boldsymbol{H}_{\{(i+1) \cdot n_t, \text{aug}\}}, \boldsymbol{H}_{\{(i+1) \cdot n_t+1, \text{aug}\}}, \ldots, \boldsymbol{H}_{\{(i+1) \cdot n_t+n_t-1, \text{aug}\}}\}$.

**Intra-sequence contrast** We hypothesize that if the Mamba Block possesses strong sequence selection capabilities, then the outputs $\{\boldsymbol{H}_{\{i \cdot n_t, \text{aug}\}}, \boldsymbol{H}_{\{i \cdot n_t+1, \text{aug}\}}, \ldots, \boldsymbol{H}_{\{i \cdot n_t+n_t-1, \text{aug}\}}\}$ of the augmented sequence at the same time step should exhibit high similarity, while ignoring progressively increasing noise. Conversely, the outputs $\boldsymbol{H}_{\{i \cdot n_t, \text{aug}\}}$ at the current time step and $\boldsymbol{H}_{\{(i+1) \cdot n_t, \text{aug}\}}$ at

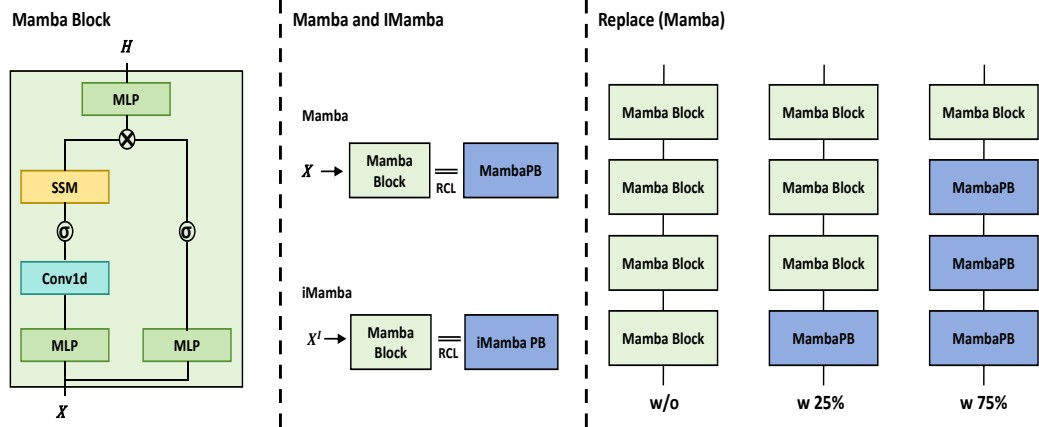

Figure 2: (a) The structure of the Mamba block. (b) Pre-training MambaPB and iMambaPB. (c) Parameter replacement during fine-tuning.

the subsequent time step should have low similarity. Therefore, we define outputs at the same time step as positive examples, while outputs at the current and subsequent time steps serve as negative examples. The objective is to minimize the distance between positive examples and maximize the distance between negative examples within the sequence, thereby enhancing the Mamba Block's sequence selection capabilities. Specifically, we use $\boldsymbol{H}_{\{i \cdot n_t, \text{aug}\}}$ as an anchor to form $n_t - 1$ positive samples and one negative sample, measuring similarity between samples using cosine similarity and employing the InfoNCE loss function Oord et al. (2018).

$$\mathcal{L}_{\text{Intra}} = -\frac{1}{s-1}\sum_{i=0}^{s-2}\frac{1}{n_t-1}\sum_{z=1}^{n_t-1}\log\frac{\exp(\text{sim}(\boldsymbol{H}_{\{i \cdot n_t, \text{aug}\}}, \boldsymbol{H}_{\{i \cdot n_t+z, \text{aug}\}})/\tau)}{\exp(\text{sim}(\boldsymbol{H}_{\{i \cdot n_t, \text{aug}\}}, \boldsymbol{H}_{\{(i+1) \cdot n_t, \text{aug}\}})/\tau)} \tag{6}$$

where $s$ is the sequence length, $i$ is the time step index, $n_t$ is the repetition count, $\tau$ is a temperature coefficient controlling the distinction of negative samples, and $\text{sim}(\cdot, \cdot)$ denotes the cosine similarity function, defined as:

$$\text{sim}(h_i, h_j) = \frac{h_i \cdot h_j}{\|h_i\|\|h_j\|} \tag{7}$$

Intra-sequence contrast allows the Mamba Block to disregard noisy, repetitive time steps while prioritizing meaningful and effective ones, thereby strengthening its selection capabilities and noise resilience.

**Inter-sequence contrast** The inter-sequence contrast further enhances contrastive learning effects while preserving selection capability and temporal correlations on the original sequence, ensuring that the Mamba Block does not overfit to augmented data. Here, $\{\boldsymbol{H}_{\{i \cdot n_t, \text{aug}\}}, \boldsymbol{H}_{\{i \cdot n_t+1, \text{aug}\}}, \dots, \boldsymbol{H}_{\{i \cdot n_t+n_t-1, \text{aug}\}}\}$ and $\boldsymbol{H}_i$ are defined as positive samples since they both represent the same time step and should maintain consistency across different time series lengths. Simultaneously, $\boldsymbol{H}_i$ and $\boldsymbol{H}_{i+1}$ are defined as negative samples to maintain selection capability on the original sequence.

$$\mathcal{L}_{\text{Inter}} = -\frac{1}{s-1}\sum_{i=0}^{s-2}\frac{1}{n_t}\sum_{z=0}^{n_t-1}\log\frac{\exp(\text{sim}(\boldsymbol{H}_i, \boldsymbol{H}_{\{i \cdot n_t+z, \text{aug}\}})/\tau)}{\exp(\text{sim}(\boldsymbol{H}_i, \boldsymbol{H}_{i+1})/\tau)} \tag{8}$$

where $s$, $i$, $n_t$, $\tau$, and $\text{sim}(\cdot, \cdot)$ are defined as above.

The overall optimization objective for Repetitive Contrastive Learning is:

$$\mathcal{L}_{\text{rc}} = \mathcal{L}_{\text{Intra}} + \mathcal{L}_{\text{Inter}} \tag{9}$$

It is noteworthy that the pre-training process for Repetitive Contrastive Learning is conducted exclusively on a single Mamba Block rather than the entire Mamba model. Even when sequence length is repeated, the memory usage and training time are typically lower than what is required for the entire model.

### 3.4 REPLACE AND INFERENCE

Based on different input sequences, we categorize Mamba blocks pre-trained with the input time series $X$ and the inverted time series $X^I$ as MambaPB and iMambaPB, respectively. These two types of Mamba blocks can be integrated into the majority of existing Mamba-based models. For example, in the Origin Mamba model Gu & Dao (2024), we substitute the parameters of the Mamba block with MambaPB. Similarly, for the iMamba model, we replace the block's parameters with iMambaPB. In the case of the TimeMachine model Ahamed & Cheng (2024), we also utilize iMambaPB for parameter substitution.

Since the pre-trained Mamba block is designed to achieve enhanced selection capabilities, we freeze the matrix $A$, which governs the Mamba's selectivity, during the inference stage. Other parameters, however, can be adjusted to suit the requirements of specific prediction tasks. Our experiments include a comparison of various parameter-freezing methods to illustrate the impact on model performance.

Additionally, Mamba-based models are often constructed with multiple stacked blocks, and selecting which blocks to replace with pre-trained parameters plays a crucial role in performance improvement. We replace the parameters of the Mamba SSM in all blocks with MambaPB and iMambaPB, and our experiments compare the results of different replacement ratios to highlight their effect on model performance.

## 4 EXPERIMENT

We conducted extensive experiments to validate the effectiveness of our method. In Section 4.1, we compare the prediction performance of various Mamba-based models—Mamba Gu & Dao (2024), iMamba, TimeMachine Ahamed & Cheng (2024), and Bi-Mamba Liang et al. (2024)—both with and without pre-trained parameters across multiple datasets: ETTh1, ETTh2, ETTm1, ETTm2, Traffic, and Electricity. In Section 4.2, we demonstrate the effectiveness of each component through ablation studies. In Section 4.3, we examine the time and memory overhead with and without pre-training. In Section 4.4, we show how RCL can enhance Mamba's selectivity by visualizing the hidden state and Delta $\Delta$. Details regarding the models, datasets, metrics, and training settings are provided in Appendix A. Additional visualization results are available in Appendix B, and detailed comparative experiments are presented in Appendix C.

### 4.1 MAIN RESULT

| | | ETTh1 | | ETTh2 | | ETTm1 | | ETTm2 | | Traffic | | Electricity | |
|---|---|---|---|---|---|---|---|---|---|---|---|---|---|
| | | MAE | MSE | MAE | MSE | MAE | MSE | MAE | MSE | MAE | MSE | MAE | MSE |
| Mamba | w/o | 0.6546 | 0.7672 | 1.4013 | 2.8442 | 0.5053 | 0.5432 | 0.5763 | 0.6008 | 0.4939 | 1.0279 | 0.4205 | 0.3863 |
| | w | 0.5974 | 0.6542 | 1.1536 | 2.0506 | 0.4798 | 0.4946 | 0.5646 | 0.5677 | 0.4604 | 0.9076 | 0.4168 | 0.3879 |
| | up-rate% | **8.7382** | **14.729** | **17.676** | **27.902** | **5.0465** | **8.9470** | **2.0302** | **5.5093** | **6.7827** | **11.704** | **0.8799** | -0.4142 |
| iMamba | w/o | 0.4987 | 0.4928 | 0.6926 | 0.9084 | 0.4316 | 0.3998 | 0.4160 | 0.3666 | 0.3234 | 0.6538 | 0.2627 | 0.1857 |
| | w | 0.4472 | 0.4278 | 0.6833 | 0.8595 | 0.3970 | 0.3669 | 0.3304 | 0.2469 | 0.2913 | 0.6003 | 0.2597 | 0.1827 |
| | up-rate% | **10.327** | **13.190** | **1.3428** | **5.3831** | **8.0167** | **8.2291** | **20.577** | **32.651** | **9.9258** | **8.1829** | **1.1420** | **1.6155** |
| TimeMachine | w/o | 0.3905 | 0.3833 | 0.3344 | 0.2911 | 0.3606 | 0.3342 | 0.2525 | 0.1746 | 0.3064 | 0.4983 | 0.2611 | 0.1872 |
| | w | 0.3869 | 0.3787 | 0.3298 | 0.2822 | 0.3458 | 0.3179 | 0.2508 | 0.1731 | 0.2991 | 0.4844 | 0.2586 | 0.1826 |
| | up-rate% | **0.9219** | **1.2001** | **1.3756** | **3.0574** | **4.1043** | **4.8773** | **0.6733** | **0.8591** | **2.3825** | **2.7895** | **0.9575** | **2.4573** |
| Bi-Mamba | w/o | 0.3948 | 0.3813 | 0.3443 | 0.2937 | 0.4680 | 0.5775 | 0.2704 | 0.1883 | 0.2786 | 0.587 | 0.2629 | 0.185 |
| | w | 0.3893 | 0.3794 | 0.3462 | 0.2955 | 0.4634 | 0.5701 | 0.2707 | 0.1857 | 0.2761 | 0.5787 | 0.2611 | 0.1818 |
| | up-rate% | **1.3931** | **0.4983** | -0.5518 | -0.6129 | **0.9829** | **1.2814** | -0.1109 | **1.3808** | **0.8973** | **1.4140** | **0.6847** | **1.7280** |

Table 1: Comparison of performance improvement by replacing parameters obtained by RCL. w/o denotes no parameter replacement, w denotes parameter replacement, and up-rate represents the improvement rate.

We validated the performance improvements brought by the parameters of the pre-trained Mamba block across multiple Mamba-based models, as shown in Table 1. By leveraging the pre-trained Mamba block parameters, the Mamba model demonstrated substantial gains across various datasets, with the Mean Squared Error (MSE) reduced by up to 27.9% and the Mean Absolute Error (MAE) improved by up to 17.7%, averaging an improvement of over 5%. For the iMamba model, the MAE showed gains of up to 20.6%, while the MSE improved by up to 32.7%, with an average performance increase exceeding 8%. These results indicate that the Mamba block parameters, refined through Repetitive Contrastive Learning, significantly enhance the predictive capabilities of the Mamba and iMamba models in time series tasks, yielding average improvements of 5% to 8%.

For the TimeMachine model, MSE improved by up to 4.88% and MAE by up to 4.10%, with an average improvement of 2%. While these gains are smaller compared to the Mamba and iMamba models, they remain noteworthy given that Bi-Mamba and TimeMachine are already state-of-the-art models for long-term sequence prediction. Achieving an additional 1% to 2% improvement solely by replacing the Mamba block parameters represents a meaningful advancement.

In summary, the parameters of the Mamba block, learned through the Repetitive Contrastive Learning method, consistently enhance the performance of various Mamba-based models. This underscores our method's efficacy in improving the sequence selection capability of the Mamba block and highlights its adaptability and potential for broad application.

## 4.2 Ablation Study

| | ETTh1 | | ETTh2 | |
|---|---|---|---|---|
| | MAE | MSE | MAE | MSE |
| w/o intra-sequence contrast | 0.636 | 0.743 | 1.351 | 2.659 |
| w/o inter-sequence contrast | 0.622 | 0.710 | 1.296 | 2.421 |
| w/o noise | 0.655 | 0.767 | 1.401 | 2.844 |
| our approach | **0.597** | **0.654** | **1.154** | **2.051** |

Table 2: Ablation results for our contrastive method settings, highlighting the effects of intra-sequence, inter-sequence, and noise augmentation components, which correspond to the three key parts of our model design.

| | ETTh1 | | ETTh2 | |
|---|---|---|---|---|
| | MAE | MSE | MAE | MSE |
| RCL w uniform noise | 0.601 | 0.664 | 1.158 | 2.060 |
| RCL w constant-intensity Gaussian noise | 0.600 | 0.660 | 1.155 | 2.059 |
| RCL w increasing-intensity Gaussian noise | **0.597** | **0.654** | **1.154** | **2.051** |

Table 3: Ablation study on the design of increasing-intensity Gaussian noise. We conducted a series of explorations examining different noise formats and their impact.

We conducted two ablation experiments to evaluate our proposed RCL method. All ablation experiments used a 4-layer Mamba as the baseline model. In the first ablation experiment, as shown in Table 2, we separately removed intra-sequence contrast, inter-sequence contrast, and noise. Removing intra-sequence contrast significantly reduced prediction performance because this contrast enhances the Mamba block's ability to select time steps and denoise. Without it, the model's ability to select time steps diminishes. Similarly, removing inter-sequence contrast also led to performance loss, as repeated time sequences can disrupt temporal consistency. The purpose of inter-sequence contrast is to maintain consistency with the temporal features of the original sequence. Without it, RCL cannot learn temporal features in broken sequences. The most significant performance drop occurred when noise was removed. Without added noise, repeated time steps are indistinguishable from the original ones, reducing task difficulty and failing to enhance the Mamba block's ability to resist noise and select time steps.

In the second ablation experiment, as shown in Table 3, we compared the effects of different types of noise on performance. Specifically, we compared uniform noise, constant-intensity Gaussian noise, and increasing-intensity Gaussian noise used in RCL. All three types of noise yielded good results, with uniform noise performing slightly worse than constant-intensity Gaussian noise, and constant-intensity Gaussian noise performing slightly worse than increasing-intensity Gaussian noise. The increasing-intensity Gaussian noise further accentuates differences between repeated time steps, increasing the difficulty of distinguishing effective information from noise, thereby enhancing pre-training performance.

## 4.3 ANALYSIS OF TIME AND MEMORY OVERHEAD

| | Memory(Unit: MB) | | | Time(Unit: S) | | |
|---|---|---|---|---|---|---|
| ETTh1 | Pretrain | Inference | Max Memory | Pretrain | Inference | Total |
| w/o | - | 11733 | 11733 | - | 1.69 | 1.71 |
| w | 13131 | 11470 | 13131 | 5 | 1.62 | 6.54 |
| Traffic | Pretrain | Inference | Max Memory | Pretrain | Inference | Total |
| w/o | - | 1602 | 1602 | - | 2.67 | 2.68 |
| w | 1994 | 1298 | 1994 | 6 | 2.54 | 8.54 |

Table 4: Peak memory consumption and average time overhead. The batch size for the ETTh1 dataset is 2000, while for the Traffic dataset it is 100.

Sequence repetition and Repetitive Contrastive Learning introduce additional memory and time overhead. To better understand the implications, we analyze the time and space complexity of the entire training process. The memory overhead for Mamba is determined by the number of blocks, $n_b$, and sequence length, $s_l$, yielding a complexity of $O(s_l n_b)$. During pre-training, only a single Mamba block is utilized, with input sequence lengths $n_t s$ and $s$, resulting in a space complexity of $O((n_t + 1)s)$. Meanwhile, the memory consumption during inference is represented as $O(s n_b)$. Table 4 details the memory consumption for Mamba training with $n_t = 3$ and $n_b = 4$ layers, illustrating that the peak memory overhead is comparable. As the number of Mamba layers increases, the memory requirement for pre-training remains significantly lower than that of the inference stage.

Due to Mamba's unique computational optimizations, the time complexity of a Mamba block is linear with respect to the sequence length $s_l$, denoted as $O(s_l)$. During pre-training, the sequence length is $n_t s$, whereas during inference, it is $s$. As such, the training time with pre-training is approximately $n_t + 1$ times longer compared to training without pre-training. Table 4 shows that when $n_t = 3$, the pre-training time consumption is about three times that of inference, which is consistent with our theoretical analysis.

## 4.4 ANALYSIS OF ENHANCED SELECTIVITY

We demonstrate that our proposed RCL effectively enhances the time step selection capability of the Mamba block by visualizing the Hidden state and Delta corresponding to the input time series of the Mamba block. The visualization results are shown in Figure 3. According to the principles of SSM, the Hidden state can be represented in a form similar to a recurrent neural network:

$$H_{t+1} = \overline{A}H_t + \overline{B}X_{t+1} \tag{10}$$

The matrix $A$ determines how historical temporal information is retained. In the Mamba block, the matrix $\overline{A}$ is determined by a fixed matrix $A$ and $\Delta$, where $A$ influences part of the historical information selection, and $\Delta$ influences another part. The visualization results indicate that without initializing with RCL parameters, the Hidden state is almost directly proportional to the input, and $\Delta$ is similarly proportional to the input. This suggests that directly training the Mamba block does not effectively retain historical information; the matrix $A$ nearly forgets all historical information, retaining only the current information as the hidden state.

In contrast, when training with initialized parameters, the Hidden state exhibits more complex representations, and $\Delta$ shows a more intricate temporal pattern. This indicates that the model learns

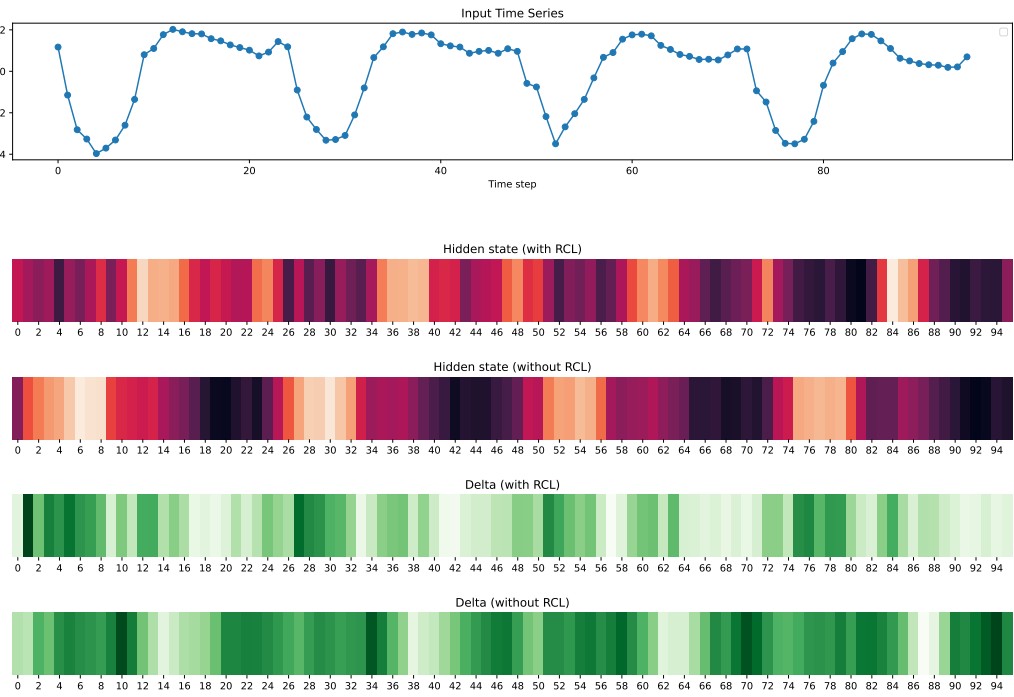

Figure 3: Visualizing the Hidden state and Delta corresponding to the input time series

complex inter-dependencies between time steps. The matrix $A$ learned by RCL demonstrates different memory and forgetting patterns for historical information across various time steps. It retains more of the input at critical time steps while preserving more historical information at non-critical time steps, thereby significantly enhancing the Mamba block's ability to select relevant information from time series data.

## 5 RELATED WORK

### 5.1 MODELS IN DEEP TIME SERIES FORECASTING

There has been extensive research focused on solving time series forecasting problems, with most works aiming to propose new models that improve prediction accuracy. Based on their model backbones, the research can be categorized into five primary groups: Transformer-based, RNN-basedHochreiter & Schmidhuber (1997), CNN-based, MLP-based, and Mamba-based models. While each of these approaches may emphasize different aspects, the key improvements revolve around addressing the specific challenges of time series tasks.

TimesNetWu et al. (2023), a CNN-based model, utilizes different periodical segmentations in both the frequency and time domains, helping models extract features from inter-period variations and infer patterns from intra-period variations. TimeMixerWang et al. (2024), which relies solely on MLP and pooling layers, outperformed all previous models by focusing on decomposing and mixing multi-scale time series data.

Transformer-based and Mamba-based models, on the other hand, mainly enhance the adaptability of their respective architectures to better address time series tasks. LogTrans Li et al. (2020) and Informer Zhou et al. (2021) introduced sparse attention mechanisms tailored for serialized data, allowing transformers to align more effectively with the nature of time series. Further advancements, such as Autoformer Wu et al. (2022) and FEDformer Zhou et al. (2022), demonstrated the critical importance of decomposing time series into seasonal and trend components. This decomposition serves as an instructive process, guiding models to better process data along the time dimension, which is essential even for attention-heavy transformer models.

PatchTST Nie et al. (2023), by segmenting time series into fixed-length patches, significantly enhances models' abilities to denoise and selectively process relevant information. Meanwhile, iTransformer Liu et al. (2024) introduces an innovative method by swapping the roles of features and time in the series, passing the time dimension through a linear layer to create 'time embeddings.' Mamba-based approaches, like TimeMachineAhamed & Cheng (2024), unify channel-mixing and channel-independence, enabling the model to effectively select the most relevant content for prediction. These methods underscore the importance of auxiliary mechanisms to enhance time series selection and processing.

## 5.2 CONTRASTIVE LEARNING

Most contrastive self-supervised learning methods have been applied primarily in the fields of visionJaiswal et al. (2021) and multimodal learningManzoor et al. (2024). This is because the objects used for contrast typically possess distinguishable high-level attributes that are easy for humans to recognize and are less susceptible to being obscured by noise. For example, image data remains interpretable to humans even when subjected to perturbations like color alterations or geometric transformations. Similarly, multimodal contrastive learning leverages cross-modality correlations, such as in visual-textual contrastive learning, where each modality provides intrinsic information to enhance the contrastive task.

In contrast, the application of contrastive learning to unimodal sequential data has been less common, often requiring specialized features. For example, CodeRetrieverLi et al. (2022) employs a similarity contrastive loss in code semantic spaces to capture nuances in code sequences. Other contrastive methods, such as those used in sequential recommendationXie & Li (2024) or text summarizationXu et al. (2022), rely on distinct sequence representations and specific training techniques to enhance contrastive performance in these domains.

In the time series domain, numerous works have focused on improving representation learning through contrastive pre-training. TS2VecYue et al. (2022) introduced a universal framework for learning time series representations at arbitrary semantic levels, emphasizing context view augmentation and hierarchical contrastive learning. Subsequently, TF-CZhang et al. (2022b) proposed a different contrastive approach by aligning time-based and frequency-based representations to achieve improved representations. Building upon this, InfoTS applied principles from information theory to prioritize high-fidelity and diverse representations, presenting a novel contrastive learning method. Meanwhile, SoftCLSLee et al. (2024) introduced soft assignments for instance-wise and temporal contrastive losses, capturing both inter-sample and intra-temporal relationships.

These methods primarily target enhanced representation learning of time series, resulting in strong performance on classification tasks but limited applicability to forecasting tasks. In contrast, our approach focuses on pre-training mamba models to capture the characteristics of recurrent noise patterns within time series. Consequently, the pre-trained parameters can be directly applied to downstream forecasting models, marking a significant point of distinction and novelty compared to existing methods.

## 6 CONCLUSION

In this paper, we propose a novel training paradigm called Repetitive Contrastive Learning (RCL) for Mamba-based models in time series tasks. The sequence selection ability of mamba block was enhanced by sequence repetition and intra-sequence and inter-sequence comparison. We conducted extensive experiments to demonstrate the efficacy of the Mamba block within the broader Mamba architecture. Our results highlight the capability of our approach in capturing the key characteristics inherent to time series data. To further substantiate our findings, we applied the method across various Mamba-based models, consistently observing significant improvements in generalization. Additionally, we evaluated resource consumption and found our method does not create additional memory burden, and the time consumption only increases linearly. Future work will aim to refine noise addition techniques and reduce training overhead to further enhance task performance.

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

Appendix

## A    BASIC INFORMATION

### A.1    MAMBA-BASED BASELINE

- **Mamba** Gu & Dao (2024): Mamba is a new Selective State Spaces model proposed by Albert Gu and Tri Dao in 2024.Li et al. (2024) It demonstrates outstanding performance in sequence modeling through its selective state space formulation, effectively capturing long-range dependencies while maintaining computational efficiency.

- **iMamba**: An enhancement of Mamba, iMamba builds upon the principles of the iTransformer, where features are treated as tokens. This model is tailored specifically for time series forecasting tasks, offering improved flexibility in feature tokenization.

- **TimeMachine** Ahamed & Cheng (2024): TimeMachine, introduced by Md Atik Ahamed and Qiang Cheng in 2024, is designed for long-term sequence forecasting. By integrating channel-independent and channel-mixed modeling approaches, it achieves state-of-the-art performance. The architecture incorporates four Mamba blocks, optimizing predictive capability over extended sequences.

- **Bi-Mamba** Liang et al. (2024): Proposed in 2024, Bi-Mamba extends the Mamba framework by adaptively capturing both internal and inter-series dependencies in multivariate time series data. The model introduces forget gates, enabling it to retain relevant historical information over extended time periods, thereby enhancing its forecasting accuracy.

### A.2    TEMPORAL BASELINE

- **Transformer**: The Transformer model, introduced by VaswaniVaswani et al. (2023) et al. in 2017, revolutionized sequence modeling by using self-attention mechanisms. Its architecture allows for efficient parallelization and effectively captures long-range dependencies, making it highly suitable for various tasks such as natural language processing and time series forecasting.

- **TimeMixer**: TimeMixerWang et al. (2024) is a novel approach designed for time series modeling, leveraging the power of mixing operations to combine temporal features. By focusing on capturing intricate temporal dependencies and interactions, TimeMixer provides robust performance in both short-term and long-term forecasting tasks.

- **CrossFormer**: CrossFormerWang et al. (2021) introduces a cross-attention mechanism specifically tailored for time series data. It excels in integrating multiple time series inputs, enabling the model to learn complex relationships across different temporal sequences, thus improving forecasting accuracy and adaptability to diverse datasets.

- **PatchTST**: PatchTSTNie et al. (2023) is a model that applies the concept of patch-based processing from computer vision to time series data. By segmenting time series into patches and processing them independently, PatchTST enhances the model's ability to capture local temporal patterns, improving efficiency and scalability for large datasets.

- **TimesNet**: TimesNetWu et al. (2023) is an advanced time series network that leverages a hierarchical structure to model temporal dependencies at multiple scales. This architecture allows TimesNet to adaptively focus on different temporal resolutions, providing superior performance in multiscale time series forecasting.

- **FEDFormer**: FEDFormer(Federated Transformer)Zhou et al. (2022) incorporates federated learning principles into the Transformer framework, allowing for decentralized time series modeling. This model is particularly effective in scenarios where data privacy is crucial, as it can learn from distributed data sources without centralizing the datasets.

- **Informer**: InformerZhou et al. (2021) is designed to efficiently handle long sequences in time series forecasting. It introduces a ProbSparse self-attention mechanism that reduces computational complexity and memory usage, making it ideal for real-time applications and large-scale datasets. Informer achieves state-of-the-art results by focusing on significant temporal patterns while filtering out noise.

## A.3 TEMPORAL PRE-TRAINING BASELINE

- **SoftCLS**: SoftCLS? is a cutting-edge model designed for contextual sequence learning. By incorporating soft clustering techniques, SoftCLS dynamically groups similar temporal patterns, enhancing the model's ability to generalize across varied contexts. This approach ensures superior performance in complex classification tasks, offering robust adaptability to fluctuating sequences while maintaining high interpretability.

- **InfoTS**: InfoTS? leverages information-theoretic principles to optimize time series modeling. By prioritizing the retention of informative features and minimizing redundancy, InfoTS significantly enhances predictive accuracy. This model excels in both supervised and unsupervised learning scenarios, making it versatile for diverse applications such as anomaly detection and trend analysis.

## A.4 DATASET

Frequency, number of features, adn time point information of the datasets.

| Dataset | Frequency | Features | Time Points | Split |
|---------|-----------|----------|-------------|-------|
| ETTh1 | Hour | 7 | 17420 | 60%/20%/20% |
| ETTh2 | Hour | 7 | 17420 | 60%/20%/20% |
| ETTm1 | 15 Minutes | 7 | 69680 | 60%/20%/20% |
| ETTm2 | 15 Minutes | 7 | 69680 | 60%/20%/20% |
| Traffic | Hour | 862 | 17544 | 60%/20%/20% |
| Electricity | Hour | 321 | 26304 | 60%/20%/20% |

## A.5 METRIC

Mean Absolute Error (MAE):

$$\text{MAE} = \frac{1}{n} \sum_{i=1}^{n} |y_i - \hat{y}_i|$$

Mean Squared Error (MSE):

$$\text{MSE} = \frac{1}{n} \sum_{i=1}^{n} (y_i - \hat{y}_i)^2$$

## A.6 MODEL SETTINGS

The parameter settings for the Mamba block during pre-training are as follows: The model dimension ($d_{model}$) is set to values [16, 32, 64], and the state dimension ($d_{state}$) is set to [16, 64, 128]. The convolution dimension ($d_{conv}$) is fixed at 4, and $pad\_vocab\_size\_multiple$ is set to 8 to ensure consistent padding sizes. The expansion factor ($expand$) is configured to 2, with $conv\_bias$ enabled (set to True) and $bias$ disabled (set to False). The repeat time, denoted as $n_t$, is set to 3, while noise variance is varied between [0.001, 0.01]. During the inference phase, the Mamba Selective State Space Model (SSM) parameters are aligned with the corresponding pre-trained block parameters to maintain consistency and leverage learned patterns effectively.

## A.7 TRAINING SETTINGS

The experiment was conducted on a server equipped with four NVIDIA GeForce RTX 3090 GPUs and an AMD EPYC 7282 16-Core Processor. During the pre-training phase, the number of layers ($n\_layer$) is set to 1, the number of epochs ($epoch$) is 100, the learning rate ($lr$) is configured to 1e-4, and the regularization coefficient is also set to 1e-4. In the inference stage, the maximum number of training epochs remains at 100, while $n\_layer$ is increased to 4. The Mean Absolute Error (MAE) serves as the loss function, and model selection is based on the lowest validation set loss. The parameter $frozentype$ is chosen as needed from the options [None, FrozenA], and the number of layers used for parameter replacement is selected from [25%, 50%, 75%, 100%], according to the

specific experimental requirements. For the prediction length, we selected four different lengths: [96, 192, 336, 720] and conducted a series of experiments. However, all results tables presented in our paper, unless otherwise specified, use a prediction length of 96. This length was chosen because it effectively illustrates the corresponding conclusions and provides a clear basis for our findings.

# B  VISUALIZATION

## B.1  VISUALIZATION OF EMBEDDING IN AUGMENTATION SEQUENCE

To visually demonstrate the impact of our contrastive learning methods, we plotted the cosine similarity values between embedding vectors of the same input sequence from the ETTm1 dataset using a heatmap Shin et al. (2006). This comparison involves identical Mamba blocks—one trained without contrastive pre-training and the other with it. The resulting variations in distribution highlight the influence of our pre-training objectives, which enhance the model's ability to selectively focus on relevant features. The images illustrate the differences in the embedding space (Figure 5) and the refined distribution achieved through contrastive learning (Figure 4).

It is evident that the Mamba model without RCL struggles to effectively distinguish between irrelevant noise and valid time steps, and it fails to make effective selections within the time series. Additionally, the original Mamba model cannot adequately separate different time steps, maintaining high correlation, which indicates that new time step information fails to be effectively encoded and merely perturbs the coding. In contrast, Mamba with RCL effectively differentiates between valid time steps and filters out noise, mitigating the effects of long sequences and introducing more valid information, thereby improving the modeling of the entire sequence.

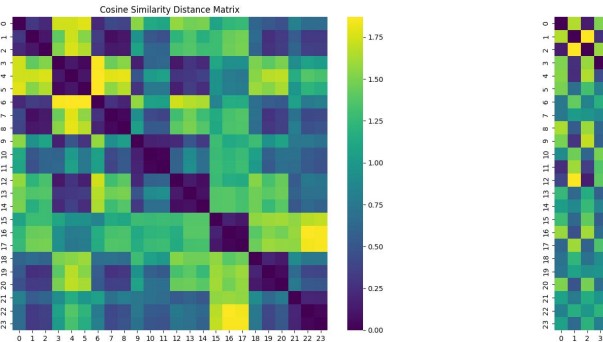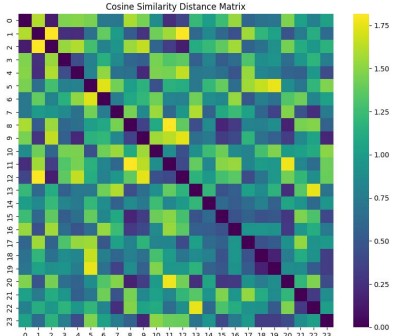

Figure 4: Visualization of model results after contrastive pre-training: the left image shows results on repeated sequences, while the right image shows results on original sequences.

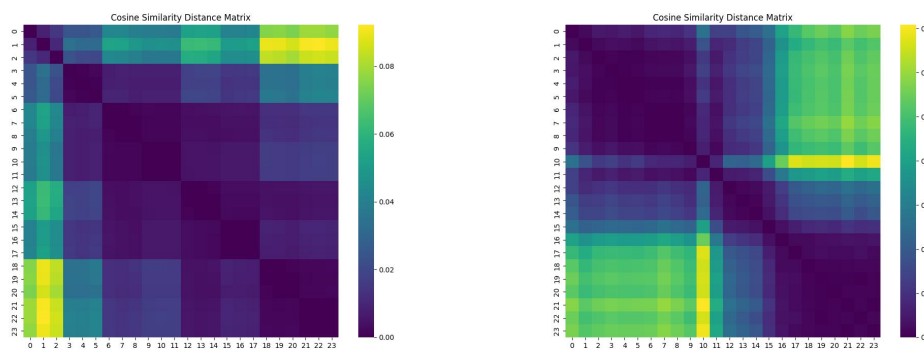

Figure 5: Visualization of model results from non-pretrained models.

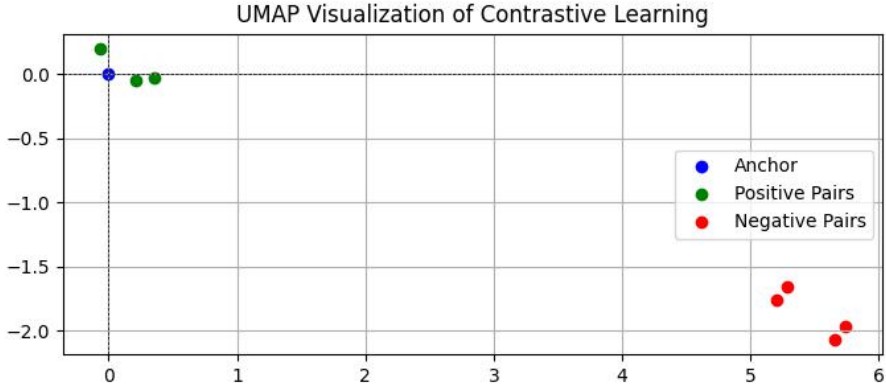

Figure 6: UMAP reduction results. Anchor points are randomly selected, and all other points are related to the anchor.

## B.2 Visualization of clustering of Positive and Negative Cases

We also visualized the detailed distribution of vectors using the UMAP technique for dimensionality reduction, where the original dimensionality of the embedding vectors is 32. UMAP is based on a theoretical framework rooted in Riemannian geometry and algebraic topology, resulting in a scalable and practical algorithm suitable for contrastive learning data McInnes et al. (2020). In the visualizations (Figure 6), we randomly selected embedding vectors from input sequences and plotted the corresponding vectors for both positive and negative pairs in our method.

The clustering results demonstrate that the model can effectively distinguish between positive and negative examples, with positive examples clustering near the anchor and negative examples retreating farther away. The significance of this distinction is evident in the clustering results, indicating that our method can better recognize valid and invalid time steps, and possesses stronger differentiation and selection capabilities.

# C  COMPARISON

## C.1  COMPARISON WITH TEMPORAL MODEL

We compared our approach with existing state-of-the-art time series prediction models, as shown in Table 5. TimeMachine* and Bi-Mamba* refer to the TimeMachine and Bi-Mamba models initialized with parameters obtained using RCL. We set all input lengths to 96 and conducted experiments across multiple prediction horizons $T = \{96, 192, 336, 720\}$. Our method achieves optimal results across various datasets and prediction horizons. For datasets with fewer data channels, our approach consistently achieves the best Mean Absolute Error (MAE) results across all prediction horizons, and Mean Squared Error (MSE) results are generally among the top two. For datasets with more channels, such as traffic and electricity, our method shows more significant improvements for longer prediction targets. This indicates enhanced stability in long-sequence predictions, attributed to the parameters obtained through RCL, which enable the Mamba block to have stronger selectivity for time series data.

| Model | | TimeMachine* | | TimeMachine | | Bi-Mamba* | | Bi-Mamba | | iTransformer | | TimeMixer | | CrossFormer | | PatchTST | | TimesNet | | FEDFormer | | Informer | |
|---|---|---|---|---|---|---|---|---|---|---|---|---|---|---|---|---|---|---|---|---|---|---|---|
| Metric | | MAE | MSE | MAE | MSE | MAE | MSE | MAE | MSE | MAE | MSE | MAE | MSE | MAE | MSE | MAE | MSE | MAE | MSE | MAE | MSE | MAE | MSE |
| ETTh1 | 96 | 0.387 | 0.379 | 0.391 | 0.383 | 0.389 | 0.379 | 0.395 | 0.381 | 0.405 | 0.386 | 0.400 | 0.375 | 0.448 | 0.423 | 0.419 | 0.414 | 0.402 | 0.384 | 0.419 | 0.376 | 0.713 | 0.865 |
| | 192 | 0.420 | 0.440 | 0.423 | 0.440 | 0.421 | 0.425 | 0.428 | 0.427 | 0.436 | 0.441 | 0.421 | 0.429 | 0.474 | 0.471 | 0.445 | 0.460 | 0.429 | 0.436 | 0.448 | 0.420 | 0.792 | 1.008 |
| | 336 | 0.442 | 0.482 | 0.446 | 0.490 | 0.456 | 0.481 | 0.459 | 0.484 | 0.458 | 0.487 | 0.458 | 0.484 | 0.546 | 0.570 | 0.466 | 0.501 | 0.469 | 0.491 | 0.465 | 0.459 | 0.809 | 1.107 |
| | 720 | 0.466 | 0.488 | 0.470 | 0.496 | 0.496 | 0.496 | 0.496 | 0.516 | 0.491 | 0.503 | 0.482 | 0.498 | 0.621 | 0.653 | 0.488 | 0.500 | 0.500 | 0.521 | 0.507 | 0.506 | 0.865 | 1.181 |
| ETTh2 | 96 | 0.330 | 0.282 | 0.334 | 0.291 | 0.347 | 0.300 | 0.349 | 0.307 | 0.349 | 0.297 | 0.341 | 0.289 | 0.584 | 0.745 | 0.348 | 0.302 | 0.374 | 0.340 | 0.397 | 0.358 | 1.525 | 3.755 |
| | 192 | 0.382 | 0.355 | 0.385 | 0.369 | 0.394 | 0.373 | 0.398 | 0.377 | 0.400 | 0.380 | 0.392 | 0.372 | 0.656 | 0.877 | 0.400 | 0.388 | 0.414 | 0.402 | 0.439 | 0.429 | 1.931 | 5.602 |
| | 336 | 0.420 | 0.412 | 0.428 | 0.421 | 0.429 | 0.434 | 0.434 | 0.435 | 0.432 | 0.428 | 0.414 | 0.386 | 0.731 | 1.043 | 0.433 | 0.426 | 0.452 | 0.452 | 0.487 | 0.496 | 1.835 | 4.721 |
| | 720 | 0.430 | 0.412 | 0.439 | 0.424 | 0.602 | 0.731 | 0.597 | 0.715 | 0.445 | 0.427 | 0.434 | 0.412 | 0.763 | 1.104 | 0.446 | 0.431 | 0.468 | 0.462 | 0.474 | 0.463 | 1.625 | 3.647 |
| ETTm1 | 96 | 0.346 | 0.318 | 0.361 | 0.334 | 0.358 | 0.332 | 0.364 | 0.332 | 0.368 | 0.334 | 0.357 | 0.320 | 0.426 | 0.404 | 0.367 | 0.329 | 0.375 | 0.338 | 0.419 | 0.379 | 0.571 | 0.672 |
| | 192 | 0.377 | 0.375 | 0.379 | 0.379 | 0.384 | 0.369 | 0.389 | 0.378 | 0.391 | 0.377 | 0.381 | 0.361 | 0.451 | 0.450 | 0.385 | 0.367 | 0.387 | 0.374 | 0.441 | 0.426 | 0.669 | 0.795 |
| | 336 | 0.387 | 0.396 | 0.394 | 0.401 | 0.404 | 0.404 | 0.412 | 0.405 | 0.420 | 0.426 | 0.404 | 0.390 | 0.515 | 0.532 | 0.410 | 0.399 | 0.411 | 0.410 | 0.459 | 0.445 | 0.871 | 1.212 |
| | 720 | 0.429 | 0.455 | 0.431 | 0.467 | 0.441 | 0.458 | 0.452 | 0.466 | 0.459 | 0.491 | 0.441 | 0.454 | 0.589 | 0.666 | 0.439 | 0.454 | 0.450 | 0.478 | 0.490 | 0.543 | 0.823 | 1.166 |
| ETTm2 | 96 | 0.251 | 0.173 | 0.253 | 0.175 | 0.271 | 0.186 | 0.270 | 0.188 | 0.264 | 0.180 | 0.258 | 0.175 | 0.366 | 0.287 | 0.259 | 0.175 | 0.267 | 0.187 | 0.287 | 0.203 | 0.453 | 0.365 |
| | 192 | 0.293 | 0.238 | 0.294 | 0.238 | 0.313 | 0.254 | 0.315 | 0.257 | 0.309 | 0.250 | 0.299 | 0.237 | 0.492 | 0.414 | 0.302 | 0.241 | 0.328 | 0.269 | 0.328 | 0.269 | 0.563 | 0.533 |
| | 336 | 0.333 | 0.299 | 0.337 | 0.307 | 0.364 | 0.316 | 0.387 | 0.392 | 0.348 | 0.311 | 0.340 | 0.298 | 0.542 | 0.597 | 0.343 | 0.305 | 0.351 | 0.321 | 0.366 | 0.325 | 0.887 | 1.363 |
| | 720 | 0.392 | 0.402 | 0.394 | 0.402 | 0.413 | 0.404 | 0.430 | 0.429 | 0.430 | 0.412 | 0.407 | 0.391 | 1.042 | 1.730 | 0.400 | 0.402 | 0.403 | 0.408 | 0.415 | 0.421 | 1.338 | 3.379 |
| Traffic | 96 | 0.299 | 0.484 | 0.306 | 0.498 | 0.276 | 0.579 | 0.279 | 0.587 | 0.268 | 0.395 | 0.285 | 0.462 | 0.290 | 0.522 | 0.359 | 0.544 | 0.321 | 0.593 | 0.366 | 0.587 | 0.368 | 0.274 |
| | 192 | 0.273 | 0.412 | 0.274 | 0.417 | 0.308 | 0.625 | 0.306 | 0.630 | 0.276 | 0.417 | 0.296 | 0.473 | 0.293 | 0.530 | 0.354 | 0.540 | 0.336 | 0.617 | 0.373 | 0.604 | 0.386 | 0.296 |
| | 336 | 0.279 | 0.429 | 0.281 | 0.433 | 0.311 | 0.666 | 0.307 | 0.659 | 0.283 | 0.433 | 0.296 | 0.498 | 0.305 | 0.558 | 0.358 | 0.551 | 0.336 | 0.629 | 0.383 | 0.621 | 0.394 | 0.300 |
| | 720 | 0.298 | 0.459 | 0.300 | 0.467 | 0.336 | 0.689 | 0.338 | 0.702 | 0.317 | 0.467 | 0.313 | 0.506 | 0.328 | 0.589 | 0.375 | 0.586 | 0.350 | 0.640 | 0.382 | 0.626 | 0.439 | 0.373 |
| Electricity | 96 | 0.259 | 0.183 | 0.261 | 0.187 | 0.261 | 0.182 | 0.263 | 0.185 | 0.240 | 0.148 | 0.247 | 0.153 | 0.314 | 0.219 | 0.285 | 0.195 | 0.272 | 0.168 | 0.308 | 0.193 | 0.391 | 0.719 |
| | 192 | 0.246 | 0.152 | 0.250 | 0.158 | 0.268 | 0.188 | 0.272 | 0.191 | 0.253 | 0.162 | 0.256 | 0.166 | 0.322 | 0.231 | 0.289 | 0.199 | 0.289 | 0.184 | 0.315 | 0.201 | 0.379 | 0.696 |
| | 336 | 0.261 | 0.169 | 0.268 | 0.172 | 0.283 | 0.200 | 0.290 | 0.212 | 0.269 | 0.178 | 0.277 | 0.185 | 0.337 | 0.246 | 0.305 | 0.215 | 0.300 | 0.198 | 0.329 | 0.214 | 0.420 | 0.777 |
| | 720 | 0.295 | 0.201 | 0.298 | 0.207 | 0.317 | 0.255 | 0.323 | 0.259 | 0.317 | 0.225 | 0.310 | 0.225 | 0.363 | 0.280 | 0.337 | 0.256 | 0.320 | 0.220 | 0.355 | 0.246 | 0.472 | 0.864 |

Table 5: Comparison results with temporal model. Bolded numbers indicate optimal results and underscores indicate sub-optimal results.

## C.2  COMPARISON WITH PRE-TRAINING METHODS

We conducted a series of experiments on the latest pre-training methods in the time series domain Luo et al. (2023); Lee et al. (2024). The results, presented in Table 6, were derived from models trained using official code on multivariate forecasting tasks. Two important aspects warrant attention. First, both methods are designed to enhance the representation learning of time series features through contrastive pre-training, heavily relying on the capabilities of feature extraction modules.

| Model | | TimeMachine* | | Bi-Mamba* | | Mamba* | | iMamba* | | InfoTS(TS2Vec) | | SoftCLS(TS2Vec) | | SoftCLS(Mamba) | | InfoTS(Mamba) | |
|---|---|---|---|---|---|---|---|---|---|---|---|---|---|---|---|---|---|
| Metric | | MAE | MSE | MAE | MSE | MAE | MSE | MAE | MSE | MAE | MSE | MAE | MSE | MAE | MSE | MAE | MSE |
| ETTh1 | 96 | 0.387 | 0.379 | 0.389 | 0.379 | 0.575 | 0.657 | 0.499 | 0.493 | 0.623 | 0.736 | 0.616 | 0.704 | 0.696 | 0.891 | 0.816 | 1.147 |
| | 192 | 0.420 | 0.440 | 0.421 | 0.425 | 0.602 | 0.713 | 0.508 | 0.532 | 0.690 | 0.857 | 0.670 | 0.810 | 0.737 | 0.959 | 0.835 | 1.186 |
| | 336 | 0.442 | 0.482 | 0.456 | 0.481 | 0.608 | 0.715 | 0.513 | 0.550 | 0.769 | 1.024 | 0.740 | 0.950 | 0.640 | 1.064 | 0.861 | 1.231 |
| ETTh2 | 96 | 0.330 | 0.282 | 0.347 | 0.300 | 1.228 | 2.124 | 0.693 | 0.908 | 0.754 | 0.936 | 0.799 | 1.015 | 0.997 | 1.542 | 0.897 | 1.219 |
| | 192 | 0.382 | 0.355 | 0.394 | 0.373 | 1.237 | 2.164 | 1.023 | 1.821 | 1.112 | 2.022 | 1.251 | 2.559 | 1.343 | 2.820 | 1.251 | 2.506 |
| | 336 | 0.420 | 0.412 | 0.429 | 0.434 | 1.234 | 2.153 | 1.073 | 2.042 | 1.264 | 2.482 | 1.312 | 2.639 | 1.402 | 2.952 | 1.327 | 2.733 |
| ETTm1 | 96 | 0.346 | 0.318 | 0.358 | 0.332 | 0.492 | 0.528 | 0.432 | 0.400 | 0.540 | 0.602 | 0.534 | 0.581 | 0.623 | 0.808 | 0.741 | 0.985 |
| | 192 | 0.377 | 0.375 | 0.384 | 0.369 | 0.513 | 0.587 | 0.450 | 0.439 | 0.575 | 0.649 | 0.569 | 0.635 | 0.654 | 0.849 | 0.756 | 1.014 |
| | 336 | 0.387 | 0.396 | 0.407 | 0.404 | 0.817 | 1.457 | 0.491 | 0.509 | 0.622 | 0.729 | 0.610 | 0.697 | 0.681 | 0.885 | 0.770 | 1.040 |
| ETTm2 | 96 | 0.251 | 0.173 | 0.271 | 0.186 | 0.576 | 0.601 | 0.416 | 0.367 | 0.452 | 0.377 | 0.460 | 0.400 | 0.491 | 0.437 | 0.782 | 0.969 |
| | 192 | 0.293 | 0.238 | 0.313 | 0.254 | 0.667 | 0.847 | 0.497 | 0.495 | 0.560 | 0.542 | 0.580 | 0.587 | 0.591 | 0.590 | 0.857 | 1.152 |
| | 336 | 0.333 | 0.299 | 0.364 | 0.316 | 0.705 | 0.922 | 0.793 | 1.032 | 0.713 | 0.846 | 0.730 | 0.885 | 0.730 | 0.855 | 0.969 | 1.461 |
| Electricity | 96 | 0.259 | 0.183 | 0.261 | 0.182 | 0.423 | 0.393 | 0.260 | 0.183 | 0.290 | 0.380 | 0.401 | 0.326 | 0.553 | 0.571 | 0.531 | 0.524 |
| | 192 | 0.246 | 0.152 | 0.270 | 0.188 | 0.430 | 0.405 | 0.280 | 0.205 | 0.293 | 0.383 | 0.403 | 0.327 | 0.555 | 0.573 | 0.532 | 0.524 |
| | 336 | 0.261 | 0.169 | 0.283 | 0.200 | 0.435 | 0.411 | 0.298 | 0.222 | 0.311 | 0.396 | 0.416 | 0.344 | 0.565 | 0.581 | 0.540 | 0.538 |

Table 6: Comparison results with pre-training methods. Bolded names with an asterisk indicate models using our pre-training methods. Parentheses following InfoTS and SoftCLS denote the backbone models utilized during pre-training. The best results for each metric are highlighted in bold.

Specifically, their experiments utilized the TSEncoder from woTS2VecYue et al. (2022) or TC from CATCCEldele et al. (2023) as feature extractors. These models are structurally distinct from mamba-based models, leading to a decline in performance when feature extraction is adapted to mamba models. Second, these approaches primarily benefit classification tasks due to their ability to accurately and effectively represent time series nodes, which aids classification but demonstrates limited improvements in forecasting tasks especially in multivariate tasks. Consequently, during their prediction stages, they use feature vectors from pre-training train a linear model to predict future values instead of leveraging pre-trained modules to construct new models. In contrast, our pre-training approach guides mamba blocks to learn sampling rules inherent in natural time sequences and identify meaningful historical information. This aligns with the requirements of forecasting tasks, allowing us to directly leverage parameters in forecasting models for superior results.

## C.3 IMPROVEMENTS UNDER DIFFERENT PARAMETERS

| Dataset | | ETTh1 | | ETTh2 | | ETTm1 | | ETTm2 | |
|---|---|---|---|---|---|---|---|---|---|
| dstate | | MAE | MSE | MAE | MSE | MAE | MSE | MAE | MSE |
| MambaDs16 | w/o | 0.6546 | 0.7672 | 1.4013 | 2.8442 | 0.5053 | 0.5432 | 0.5763 | 0.6008 |
| | w | 0.5974 | 0.6542 | 1.1536 | 2.0506 | 0.4798 | 0.4946 | 0.5646 | 0.5677 |
| | up-rate% | **8.7382** | **14.7289** | **17.6764** | **27.9024** | **5.0465** | **8.9470** | **2.0302** | **5.5093** |
| MambaDs32 | w/o | 0.6394 | 0.7359 | 1.2478 | 2.3706 | 0.5246 | 0.5729 | 0.6106 | 0.6778 |
| | w | 0.5741 | 0.6369 | 1.2137 | 2.2131 | 0.4958 | 0.5182 | 0.5199 | 0.5106 |
| | up-rate% | **10.2127** | **13.4529** | **2.7328** | **6.6439** | **5.4899** | **9.5479** | **14.8542** | **24.6680** |
| MambaDs64 | w/o | 0.6382 | 0.7424 | 1.1759 | 2.1532 | 0.4961 | 0.5562 | 0.5803 | 0.5379 |
| | w | 0.6247 | 0.7280 | 1.1323 | 2.0257 | 0.4828 | 0.5124 | 0.5771 | 0.5289 |
| | up-rate% | **2.1153** | **1.9397** | **3.7078** | **5.9214** | **2.6809** | **7.8749** | **0.5514** | **1.6732** |

Table 7: Comparison of Mamba's performance under different dstate. Mamba-DsN represents Mamba with $dstate$ set to N.

| Dataset | | ETTh1 | | ETTh2 | | ETTm1 | | ETTm2 | | Traffic | |
|---|---|---|---|---|---|---|---|---|---|---|---|
| dmodel | | MAE | MSE | MAE | MSE | MAE | MSE | MAE | MSE | MAE | MSE |
| MambaDm32 | w/o | 0.6096 | 0.7107 | 1.2655 | 2.6822 | 0.7123 | 0.9830 | 0.6630 | 0.7246 | 0.4959 | 1.0047 |
| | w | 0.5993 | 0.6707 | 1.0751 | 2.0090 | 0.6628 | 0.9166 | 0.5739 | 0.6602 | 0.4865 | 0.9621 |
| | up-rate% | **1.6896** | **5.6283** | **15.0454** | **25.0988** | **6.9493** | **6.7548** | **13.4389** | **8.8877** | **1.8955** | **4.2401** |
| MambaDm64 | w/o | 0.6243 | 0.7188 | 1.2142 | 2.2030 | 0.7026 | 1.0190 | 0.5793 | 0.6063 | 0.3347 | 0.6652 |
| | w | 0.6000 | 0.6778 | 1.1267 | 1.8989 | 0.6959 | 0.9642 | 0.5580 | 0.5635 | 0.2976 | 0.5976 |
| | up-rate% | **3.8924** | **5.7040** | **7.2064** | **13.8039** | **0.9536** | **5.3778** | **3.6769** | **7.0592** | **11.0846** | **10.1624** |
| MambaDm128 | w/o | 0.6546 | 0.7672 | 1.4013 | 2.8442 | 0.5053 | 0.5432 | 0.5763 | 0.6008 | 0.4791 | 0.9903 |
| | w | 0.5974 | 0.6542 | 1.1536 | 2.0506 | 0.4798 | 0.4946 | 0.5646 | 0.5677 | 0.4622 | 0.9467 |
| | up-rate% | **8.7382** | **14.7289** | **17.6764** | **27.9024** | **5.0465** | **8.9470** | **2.0302** | **5.5093** | **3.5274** | **4.4027** |

Table 8: Comparison of Mamba's performance under different dmodel. Mamba-DmN represents Mamba with $dmodel$ set to N.

To further illustrate the generality of our method, we evaluated its enhancement capabilities on the Mamba model with varying parameters. Specifically, we standardized all Mamba models to contain four Mamba blocks and compared their performance under different values of $d_{state}$ and $d\_model$. Table 7 presents a performance comparison of the model across various $d_{state}$ values. While Mamba's performance varied depending on $d_{state}$ for the same dataset, our method consistently delivered improvements, achieving an average increase of over 5% and a maximum gain of 24.67%. Similarly, we assessed the improvement effects of our method under different values of $d_{model}$, as detailed in Table 8. Across multiple datasets, every tested $d_{model}$ exhibited significant improvements, with a maximum gain of 27.9% and an average increase of 6%.

These experiments demonstrate that our method is robust and unaffected by specific model parameters, consistently providing performance enhancements regardless of parameter variations. Results across diverse datasets and parameter settings reinforce this conclusion. Furthermore, within the same dataset, our method effectively narrows performance gaps caused by parameter variations, aligning results closer to optimal performance. This not only boosts the stability and robustness of Mamba-based models but also reduces the time and effort required for parameter tuning.

## C.4 Comparison of Replacement and Freezing Methods

| | | ETTm1 | | | | ETTm2 | | | |
| | | None | | FrozenA | | None | | FrozenA | |
| | | MAE | MSE | MAE | MSE | MAE | MSE | MAE | MSE |
|---|---|---|---|---|---|---|---|---|---|
| | w/o | 0.5053 | 0.5432 | 0.5053 | 0.5432 | 0.5763 | 0.6008 | 0.5763 | 0.6008 |
| layer-25% | w | 0.4921 | 0.5394 | 0.4921 | 0.5393 | 0.6609 | 0.7902 | 0.5611 | 0.5696 |
| | up-rate% | **2.6123** | **0.6996** | **2.6123** | **0.7180** | -14.6799 | -31.5246 | **2.6375** | **5.1931** |
| layer-50% | w | 0.4798 | 0.4946 | 0.4976 | 0.5548 | 0.6021 | 0.6230 | 0.6389 | 0.7423 |
| | up-rate% | **5.0465** | **8.9470** | **1.5238** | -2.1355 | -4.4768 | -3.6951 | -10.8624 | -23.5519 |
| layer-75% | w | 0.4816 | 0.5256 | 0.4816 | 0.5255 | 0.5299 | 0.5366 | 0.5646 | 0.5676 |
| | up-rate% | **4.6903** | **3.2401** | **4.6903** | **3.2585** | **8.0514** | **10.6858** | **2.0302** | **5.5260** |
| layer-100% | w | 0.5106 | 0.5692 | 0.5016 | 0.5658 | 0.5486 | 0.5735 | 0.5296 | 0.5258 |
| | up-rate% | -1.0489 | -4.7865 | **0.7322** | -4.1605 | **4.8065** | **4.5439** | **8.1034** | **12.4834** |

Table 9: Comparison of Replacement and Freezing Methods. The "layer-x%" indicates that the first x% of layers were replaced by pre-trained blocks.

A Mamba-based model typically comprises multiple Mamba blocks. Each Mamba block contains a matrix $A$, which is defined in 3.2. The parameters are responsible for controlling the block's selectivity towards information before. To evaluate the impact of parameter replacement and parameter freezing during the inference stage, we used a 4-layer Mamba model as a baseline. The replacement strategy involved substituting 25%, 50%, 75%, and 100% of the Mamba blocks, while the parameter freezing strategy was categorized into no freezing (None) and freezing of matrix $A$ (FrozenA). Freezing matrix $A$ helps preserve the enhanced selectivity gained during pre-training.

As shown in Table 9, the optimal parameter replacement and freezing strategies differ across datasets. For the ETTm1 dataset, replacing 50% of the Mamba blocks without freezing any parameters yielded the greatest improvement, while replacing 100% of the blocks resulted in the lowest performance. This suggests that the selection capabilities of the pre-trained parameters do not fully align with the prediction target. By replacing only 50% of the Mamba blocks, the model can better encode the time series, while the remaining blocks focus on fitting the specific prediction requirements of the dataset, ultimately enhancing model performance.

Conversely, for the ETTm2 dataset, the greatest improvement was achieved by replacing all Mamba blocks and freezing matrix $A$. In this case, the selective enhancements from pre-training aligned well with the dataset's prediction targets. This approach preserved the pre-trained parameters' selectivity while allowing the remaining parameters to adjust to fit the prediction targets effectively.

Similar results were observed across other datasets. Broadly, the findings can be grouped into two effective strategies: replacing 50% of the Mamba blocks without freezing any parameters and replacing 100% of the Mamba blocks while freezing matrix $A$. We recommend choosing between these two approaches during the inference phase for optimal performance.

## C.5 Detail Comparison of Improvements

To demonstrate that pre-training Mamba blocks with RCL can effectively enhance the temporal prediction capabilities of Mamba-based models, we present the performance improvements of four Mamba-based models after using pre-trained parameters. We conducted extensive testing on six datasets, each with an input length of 96 and prediction lengths of $\{96, 192, 336, 720\}$. To clearly illustrate the performance improvements, we provide the percentage increase in MSE and MAE when using pre-trained parameters compared to not using them, as shown by the up-rate in Table 10.

The results indicate that, for the vast majority of datasets and prediction lengths, the parameters obtained through our method enhance the predictive performance of Mamba-based models, demonstrating that our approach is generally effective. By pre-training a Mamba block and using the pre-trained parameters to initialize all mamba blocks in Mamba-based model, the original model's temporal prediction performance can be significantly improved.

| | | | ETTh1 | | ETTh2 | | ETTm1 | | ETTm2 | | Traffic | | Electricity | |
|---|---|---|---|---|---|---|---|---|---|---|---|---|---|---|
| | | | MAE | MSE | MAE | MSE | MAE | MSE | MAE | MSE | MAE | MSE | MAE | MSE |
| Mamba | 96 | w/o | 0.6546 | 0.7672 | 1.4013 | 2.8442 | 0.5053 | 0.5432 | 0.5763 | 0.6008 | 0.4939 | 1.0279 | 0.4205 | 0.3863 |
| | | w | 0.5974 | 0.6542 | 1.1536 | 2.0506 | 0.4798 | 0.4946 | 0.5646 | 0.5677 | 0.4604 | 0.9076 | 0.4168 | 0.3879 |
| | | up-rate% | **8.7382** | **14.729** | **17.676** | **27.902** | **5.0465** | **8.9470** | **2.0302** | **5.5093** | **6.7827** | **11.704** | **0.8799** | **-0.4142** |
| | 192 | w/o | 0.6298 | 0.7115 | 1.2371 | 2.1642 | 0.5126 | 0.5866 | 0.6670 | 0.8471 | 0.5617 | 1.1962 | 0.4298 | 0.4053 |
| | | w | 0.6021 | 0.7127 | 1.0509 | 1.9490 | 0.4970 | 0.5524 | 0.5655 | 0.5573 | 0.5610 | 1.1877 | 0.4288 | 0.4130 |
| | | up-rate% | **4.3982** | **-0.1687** | **15.0513** | **9.9436** | **3.0433** | **5.8302** | **15.2174** | **34.2108** | **0.1246** | **0.7106** | **0.2327** | **-1.8998** |
| | 336 | w/o | 0.6383 | 0.7210 | 1.2341 | 2.1528 | 0.8172 | 1.4569 | 0.7052 | 0.9220 | 0.6025 | 1.3079 | 0.4354 | 0.4108 |
| | | w | 0.6084 | 0.7145 | 1.0497 | 1.9485 | 0.8008 | 1.4479 | 0.6270 | 0.6842 | 0.5848 | 1.2560 | 0.4324 | 0.4176 |
| | | up-rate% | **4.6843** | **0.9015** | **14.9421** | **9.4900** | **2.0069** | **0.6178** | **11.0891** | **25.7918** | **2.9378** | **3.9682** | **0.6890** | **-1.6553** |
| | 720 | w/o | 0.6776 | 0.7727 | 1.2206 | 2.1005 | 0.8235 | 1.4557 | 0.7374 | 0.9942 | 0.4893 | 1.0108 | 0.4529 | 0.4326 |
| | | w | 0.6461 | 0.7556 | 1.0541 | 1.9537 | 0.8142 | 1.4588 | 0.6682 | 0.7811 | 0.4645 | 0.9189 | 0.4447 | 0.4320 |
| | | up-rate% | **4.6488** | **2.2130** | **13.6408** | **6.9888** | **1.1293** | **-0.2130** | **9.3843** | **21.4343** | **5.0685** | **9.0918** | **1.8106** | **0.1387** |
| iMamba | 96 | w/o | 0.4987 | 0.4928 | 0.6926 | 0.9084 | 0.4316 | 0.3998 | 0.4160 | 0.3666 | 0.3234 | 0.6538 | 0.2627 | 0.1857 |
| | | w | 0.4472 | 0.4278 | 0.6833 | 0.8595 | 0.3970 | 0.3669 | 0.3304 | 0.2469 | 0.2913 | 0.6003 | 0.2597 | 0.1827 |
| | | up-rate% | **10.3268** | **13.1899** | **1.3428** | **5.3831** | **8.0167** | **8.2291** | **20.5769** | **32.6514** | **9.9258** | **8.1829** | **1.1420** | **1.6155** |
| | 192 | w/o | 0.5075 | 0.5320 | 1.0228 | 1.8207 | 0.4500 | 0.4390 | 0.4973 | 0.4949 | 0.3129 | 0.6354 | 0.2801 | 0.2047 |
| | | w | 0.4871 | 0.5143 | 0.9430 | 1.5825 | 0.4356 | 0.4174 | 0.4763 | 0.4557 | 0.3091 | 0.6335 | 0.2788 | 0.2025 |
| | | up-rate% | **4.0197** | **3.3271** | **7.8021** | **13.0829** | **3.2000** | **4.9203** | **4.2228** | **7.9208** | **1.2144** | **0.2990** | **0.4641** | **1.0747** |
| | 336 | w/o | 0.5125 | 0.5498 | 1.0727 | 2.0417 | 0.4909 | 0.5085 | 0.7932 | 1.0322 | 0.3233 | 0.6605 | 0.2987 | 0.2238 |
| | | w | 0.4750 | 0.4992 | 0.9913 | 1.7052 | 0.4677 | 0.4998 | 0.5854 | 0.6272 | 0.3216 | 0.6645 | 0.2975 | 0.2222 |
| | | up-rate% | **7.3171** | **9.2033** | **7.5883** | **16.4814** | **4.7260** | **1.7109** | **26.1977** | **39.2366** | **0.5258** | **-0.6056** | **0.4017** | **0.7149** |
| | 720 | w/o | 0.5418 | 0.5818 | 1.0534 | 1.8199 | 0.6238 | 0.7306 | 1.0698 | 2.0298 | 0.3486 | 0.7105 | 0.3342 | 0.2683 |
| | | w | 0.5391 | 0.5640 | 1.0172 | 1.7220 | 0.5120 | 0.5534 | 0.9936 | 1.5644 | 0.3475 | 0.7172 | 0.3323 | 0.2627 |
| | | up-rate% | **0.4983** | **3.0595** | **3.4365** | **5.3794** | **17.9224** | **24.2540** | **7.1228** | **22.9284** | **0.3155** | **-0.9430** | **0.5685** | **2.0872** |
| TimeMachine | 96 | w/o | 0.3905 | 0.3833 | 0.3344 | 0.2911 | 0.3606 | 0.3342 | 0.2525 | 0.1746 | 0.3064 | 0.4983 | 0.2611 | 0.1872 |
| | | w | 0.3869 | 0.3787 | 0.3298 | 0.2822 | 0.3458 | 0.3179 | 0.2508 | 0.1731 | 0.2991 | 0.4844 | 0.2586 | 0.1826 |
| | | up-rate% | **0.9219** | **1.2001** | **1.3756** | **3.0574** | **4.1043** | **4.8773** | **0.6733** | **0.8591** | **2.3825** | **2.7895** | **0.9575** | **2.4573** |
| | 192 | w/o | 0.4225 | 0.4401 | 0.3851 | 0.3685 | 0.3785 | 0.3787 | 0.2941 | 0.2381 | 0.2740 | 0.4170 | 0.2500 | 0.1580 |
| | | w | 0.4202 | 0.4399 | 0.3821 | 0.3551 | 0.3770 | 0.3750 | 0.2930 | 0.2381 | 0.2732 | 0.4115 | 0.2460 | 0.1520 |
| | | up-rate% | **0.5444** | **0.0454** | **0.7790** | **3.6364** | **0.3963** | **0.9770** | **0.3740** | **0.0000** | **0.2920** | **1.3189** | **1.6000** | **3.7975** |
| | 336 | w/o | 0.4458 | 0.4902 | 0.4281 | 0.4206 | 0.3937 | 0.4010 | 0.3371 | 0.3066 | 0.2810 | 0.4330 | 0.2680 | 0.1720 |
| | | w | 0.4419 | 0.4824 | 0.4201 | 0.4119 | 0.3867 | 0.3956 | 0.3327 | 0.2991 | 0.2790 | 0.4290 | 0.2610 | 0.1690 |
| | | up-rate% | **0.8748** | **1.5912** | **1.8687** | **2.0685** | **1.7780** | **1.3466** | **1.3053** | **2.4462** | **0.7117** | **0.9238** | **2.6119** | **1.7442** |
| | 720 | w/o | 0.4702 | 0.4959 | 0.4386 | 0.4243 | 0.4310 | 0.4670 | 0.3940 | 0.4073 | 0.3000 | 0.4670 | 0.2980 | 0.2070 |
| | | w | 0.4656 | 0.4883 | 0.4295 | 0.4119 | 0.4291 | 0.4552 | 0.3920 | 0.4018 | 0.2980 | 0.4590 | 0.2950 | 0.2010 |
| | | up-rate% | **0.9783** | **1.5326** | **2.0748** | **2.9225** | **0.4408** | **2.5268** | **0.5076** | **1.3504** | **0.6667** | **1.7131** | **1.0067** | **2.8986** |
| Bi-Mamba | 96 | w/o | 0.3948 | 0.3813 | 0.3443 | 0.2937 | 0.3641 | 0.3319 | 0.2704 | 0.1883 | 0.2786 | 0.587 | 0.2629 | 0.185 |
| | | w | 0.3893 | 0.3794 | 0.3462 | 0.2955 | 0.3578 | 0.3316 | 0.2707 | 0.1857 | 0.2761 | 0.5787 | 0.2611 | 0.1818 |
| | | up-rate% | **1.3931** | **0.4983** | **1.7303** | **0.0904** | **0.9829** | **1.2814** | **-0.1109** | **1.3808** | **0.8973** | **1.4140** | **0.6847** | **1.7280** |
| | 192 | w/o | 0.4280 | 0.4270 | 0.3977 | 0.3772 | 0.3894 | 0.3780 | 0.3145 | 0.2572 | 0.3057 | 0.6301 | 0.2715 | 0.1914 |
| | | w | 0.4210 | 0.4250 | 0.3935 | 0.3733 | 0.3840 | 0.3692 | 0.3131 | 0.2544 | 0.3081 | 0.6250 | 0.2698 | 0.1881 |
| | | up-rate% | **1.6355** | **0.4684** | **1.0561** | **1.0339** | **1.3867** | **2.3280** | **0.4452** | **1.0886** | **-0.7851** | **0.8094** | **0.6262** | **1.7241** |
| | 336 | w/o | 0.4593 | 0.4838 | 0.4340 | 0.4354 | 0.4119 | 0.4045 | 0.3871 | 0.3915 | 0.3068 | 0.6585 | 0.2896 | 0.2117 |
| | | w | 0.4563 | 0.4805 | 0.4286 | 0.4344 | 0.4069 | 0.4036 | 0.3644 | 0.3158 | 0.3107 | 0.6659 | 0.2831 | 0.1999 |
| | | up-rate% | **0.6532** | **0.6821** | **1.2442** | **0.2297** | **1.2139** | **0.2225** | **5.8641** | **19.3359** | **-1.2712** | **-1.1238** | **2.2445** | **5.5739** |
| | 720 | w/o | 0.4963 | 0.5164 | 0.5970 | 0.7150 | 0.4517 | 0.4659 | 0.4300 | 0.4292 | 0.3384 | 0.7015 | 0.3228 | 0.2591 |
| | | w | 0.4960 | 0.4962 | 0.6020 | 0.7310 | 0.4413 | 0.4579 | 0.4131 | 0.4044 | 0.3364 | 0.6894 | 0.3174 | 0.2547 |
| | | up-rate% | **0.0604** | **3.9117** | **-0.8375** | **-2.2378** | **2.3024** | **1.7171** | **3.9302** | **5.7782** | **0.5910** | **1.7249** | **1.6729** | **1.6982** |

Table 10: Detail Comparison of performance improvement by replacing parameters obtained by RCL. w/o denotes no parameter replacement, w denotes parameter replacement, and up-rate represents the improvement rate.

