# OpenReview forum: "Repetitive Contrastive Learning Enhances Mamba's Selectivity in Time Series Prediction"
_ICLR.cc/2025/Conference — Submitted to ICLR 2025_

### Official Review · Reviewer_3whs · 2024-10-31

**Soundness:** 2
**Presentation:** 2
**Contribution:** 2
**Rating:** 6
**Confidence:** 3

**Summary:**

This paper proposes a contrastive learning training framework for Mamba-based models in multivariate time series forecasting (MTSF) tasks. The authors introduce a repeated data augmentation and contrastive learning approach for both inter- and intra-sequence levels. Experiments demonstrate that replacing partial blocks of Mamba models with the proposed pre-trained blocks can enhance their performance.

**Strengths:**

1. This is one of the first works exploring contrastive learning in Mamba models for time series.
2. The proposed method demonstrates performance gains on recent Mamba-based models for time series forecasting.
3. Based on the visualizations in Figures 3 and 4, I believe the pretraining is effective.

**Weaknesses:**

1. The repetitive augmentation does not seem novel, as it appears to simply create negative pairs and concatenate positive and negative pairs together. This approach likely increases the input size and, consequently, the computational cost. Traditional contrastive learning can also generate negative pairs with different strategies, and I suspect that with a smaller input size, the model might be even more efficient. (Please correct me if I am mistaken.)

2. The paper does not include comparisons with non-Mamba methods. This is crucial, as improvements over weak baselines alone may not provide strong evidence of the proposed method’s contribution.

3. The experimental evaluation may be insufficient for ICLR standards. Recent works often include either a larger variety of datasets or more comprehensive experimental setups.

Minors: There are some typos and repeated sentences, and the overall presentation could be improved.

**Questions:**

1, Could you clarify whether all MambaPB blocks share the same parameters, or are they trained independently during the self-training phase? If all MambaPB blocks share the same parameters, this would make the approach more interesting.

2. In what way does the proposed augmentation strategy improve upon conventional contrastive learning approaches?

3. Is there an ablation study that examines how varying levels of Gaussian noise impact performance?

---

> ### Author Response · Authors · 2024-11-19
>
> Thank you for your review comments. We have added extensive experiments and included them in the Appendix. We will address your questions based on the experimental results.
>
> W1: Token-level repetition and token-level contrast are not common. Our method, RCL, is designed to pre-train a Mamba Block to achieve better initialization parameters. We apply these initialization parameters to all Mamba-based models as the initialization parameters for each Mamba Block within them. Therefore, we only pre-train a single Mamba Block, which is much smaller than the parameter size of a full model. Experimental results show that even with threefold repetition, it does not lead to excessive memory usage. In terms of time, the parameters from a single pre-training can be applied to the initialization of all Mamba-based models, thus the time cost is minimal.
>
> W2: We have provided detailed comparison results with non-Mamba models in Appendix C.1. Additionally, we have included comparisons with other pre-training methods in Appendix C.2.
>
> W3: We have also supplemented with comprehensive experiments, such as ablation studies (Appendix C.3), comparisons and improvement experiments across multiple prediction lengths (Appendix C.4), and evidence and visualization of selective enhancement (Appendix D).
>
> W4: We have revised the wording of the article in hopes of meeting your standards.
>
> Q1: All Mamba Blocks share the same set of initialization parameters. This means that on the same dataset, a single RCL pre-training is used for the parameter initialization of all Mamba Blocks in the model, and they share the same parameters.
>
> Q2: 1. Our method involves token-level contrast rather than the more common sequence-level contrast. 2. Our design perspective is similar to that of RNNs, which differs from the common attention-based contrast. We aim to enhance the selection between the current time step and historical information. If a time step is invalid, more historical information should be retained; if a time step is important, more of the current time step should be retained while forgetting historical information. Matrix A is crucial in this process. 3. Our method is targeted at the Mamba Block, and the results of a single pre-training are effective for all Mamba-based models, as validated by experiments. To my knowledge, contrastive learning specifically for the Mamba structure and parameters is rare.
>
> Q3: We have added two ablation experiments in Appendix C.3, conducting ablation studies on RCL and noise types separately.

---

> > ### Comment · Reviewer_3whs · 2024-11-21
> >
> > Thanks for your detailed response. I think the manuscript has improved, especially the experiment part. However, I agree with some comments from other reviews that the contribution may not be sufficient, and some statements are not convincing. So, I decided to keep my score, but I will keep looking at other reviewers' responses to adjust my opinion.

---

### Official Review · Reviewer_UGa7 · 2024-11-01

**Soundness:** 2
**Presentation:** 1
**Contribution:** 1
**Rating:** 5
**Confidence:** 4

**Summary:**

The paper under review presents an innovative approach to enhance the sequence selection capability of Mamba-based models in time series prediction tasks. The authors propose a method called Repetitive Contrastive Learning (RCL), which involves augmenting time series data by duplicating time points and introducing Gaussian noise. The RCL method aims to improve the model's ability to selectively focus on relevant moments within time series data, thereby enhancing its predictive performance. The experiments conducted demonstrate some improvements in performance across various Mamba-based models.

**Strengths:**

The paper tries to address a crucial gap in time series analysis by teaching models to select and prioritize key data points, which is vital for improving prediction accuracy. The authors are acknowledged for bringing attention to an issue that has often been overlooked in previous studies.

**Weaknesses:**

1. The paper suffers from poor writing quality，reducing its readability and comprehension. Symbols are ill-defined, and the operational flowchart lacks clear descriptions, hindering reader understanding. A revision to enhance language and clarity is needed.
2. Duplicating time points and adding noise may result in significant numerical differences between the replicated samples and the original samples, which could reduce their similarity. At the same time, the temporal correlations in the original data are disrupted. Why not use patch operations instead?
3. It is unreasonable for $X_{aug}$ to be input as a whole into the model because it contains all the duplicated time steps and enhanced noise. Directly passing it through an MLP may amplify the noise and completely disrupt the temporal correlations in the original data.
4. There are too few baseline comparisons, and they are not representative. Additionally, there is a lack of comparison with non-Mamba methods, making it difficult to demonstrate the superiority of the proposed approach.
5. The technical contribution is insufficient. The conclusions are inadequate as the experiments do not delve deeply into the core claims of the paper, such as whether the model actually achieves effective selection and prioritization of key moments in time series data as stated.

**Questions:**

1. Please provide a more in-depth analysis to demonstrate whether the model has indeed effectively selected and prioritized key moments in time series data, as claimed in the paper.
2. Offer results for more baselines, such as additional Mamba-based and non-Mamba-based methods.

---

> ### Author Response · Authors · 2024-11-19
>
> Thank you for your review comments. We have added a substantial number of experiments and included them in the Appendix. We will address your questions one by one based on the experimental results.
>
> Q1: In Appendix D.1, we demonstrate selective enhancement through analysis and visualization. Matrix A determines the memory and forgetting of historical information. Visualization shows that the initialization parameters learned using RCL can capture more complex temporal patterns, allowing for more flexible and complex memory and forgetting of historical information, highlighting key time steps.
>
> Q2: In Appendix C.1, we have supplemented the experimental results and analysis comparing with recent baselines. We compared the RCL-enhanced TimeMachine* and Bi-mamba*.
>
> W1: We have corrected the wording and symbol errors in the paper.
>
> W2: We considered significant numerical differences and thus used cosine similarity in the contrastive learning process to evaluate the similarity between two encodings, as this similarity is not affected by the magnitude of the values. Repeated sequences can indeed disrupt temporal correlations, but the purpose of inter-sequence contrast is to maintain consistency with the temporal correlations of the original sequence while performing intra-sequence contrast.
>
> W3: The goal of intra-sequence contrast is for the Mamba Block to distinguish between noise points and valid information. From the RNN perspective of Mamba, \\( H_{t+1} = AH_t + BX_{t+1} \\). For repeated time steps \\( X_{t,0}, X_{t,1}, X_{t,2} \\), our pre-training objective is for \\( H_{t,0}, H_{t,1}, H_{t,2} \\) to be as similar as possible. This means that during the process \\( H_{t,1} = AH_{t,0} + BX_{t,1} \\), A should retain more historical information while B can ignore the current noise step. Regarding temporal correlations, as mentioned in W2, inter-sequence contrast constrains the temporal correlation with the original sequence.
>
> W4,5: As mentioned in Q1 and Q2, we have provided more detailed experimental results in the Appendix.
>
> In addition, we have included experiments with multiple perspectives, as seen in Appendix C and D.

---

### Official Review · Reviewer_NrLR · 2024-11-03

**Soundness:** 2
**Presentation:** 1
**Contribution:** 2
**Rating:** 5
**Confidence:** 3

**Summary:**

The paper claims that it addresses the challenge of long-sequence prediction in time series forecasting by enhancing Mamba’s sequence selection capability. It introduces Repetitive Contrastive Learning with sequence augmentation and noise to improve performance, which is shown to universally benefit Mamba-based models without extra memory overhead.

**Strengths:**

1. The proposed repeating sequence augmentation and repetitive contrastive learning methods are simple, easy to implement, and transferable to various TSF model architectures.

2. Extensive experiments demonstrate that Mamba-based TSF models show significant performance improvement after applying the proposed pretraining and augmentation methods.

**Weaknesses:**

1. The paper contains some formatting errors, such as many mathematical symbols in Figures 1 and 2 not rendering correctly. The citation format is also inconsistent, with a lot of textual citations used in places where parenthetical citations would be more appropriate, making the paper harder to read.

2. About selection ability:

The paper claims that Mamba is chosen to be used in the paper because of its stronger temporal selection ability compared to previous architectures. However, vanilla attention, with its temporal-adaptive dynamical parameters, is clearly more suitable for selection task than Mamba. The improvement of Mamba's selective SSM in selection ability is compared against the fixed temporal weights of S4, not against the more flexible softmax attention.

Mamba’s selective SSM was designed to introduce selection capabilities, like the induction heads learned in Transformer-based LLMs, to fixed SSM-weighted models like S4. Due to time complexity constraints, Mamba is less likely to surpass standard attention, which has $O(L^2)$ complexity because it calculates unique softmax temporal weights for each token, maximizing selection and matching abilities. In contrast, near-linear complexity methods (like Mamba and linear attention variants) avoid computing unique weights for each token, inheriting attention weights from previous steps, making adjustments through methods like gating mechanisms rather than recalculating all the weights independently for each step, to decrease the time complexity. This temporal propagation of weights limits their ability to achieve the distinct selection ability of softmax attention for each token.

The paper’s argument that “Mamba's better performance in TSF comes from its selection ability, thus requiring better pretraining methods to enhance this ability for the performance improvement of downstream forecasting” lacks sufficient support. While the proposed contrastive pretraining method does improve performance, is this really due to enhanced selection ability? Contrastive learning literature more often emphasizes the maintenance of a well-structured latent space to distinguish samples. The performance improvement might stem from learning more distinguishable representations in the latent space rather than an unquantified enhancement in selection ability.

In summary, the paper would benefit from more discussion, analysis, and experiments to demonstrate that the proposed methods enhance the model's selective ability. For instance, the authors could adopt the analysis techniques from LLM literature that show when and how induction heads are learned during training to compare the selective ability before and after applying the methods in this paper. Otherwise, the paper may need rebranding to explain the performance improvement from alternative perspectives, such as emphasizing how contrastive pretraining enhances the geometric properties of the latent space to improve the representation and discrimination of time series, as suggested in prior contrastive learning literature.

**Questions:**

1. The proposed method seems not to be specific to the Mamba structure. Applying it to more structures like Transformer could help the community better understand its significance.

2. The paper lacks comparisons with existing time series pretraining and representation learning methods. Since the approach introduces a pre-training / data-augmentation stage rather than being end-to-end, fair comparisons with previous time series representation learning / data augmentation + downstream forecasting methods are needed.

3. Mamba is inherently a sequential SSM for ordered input tokens. It seems the authors used vanilla Mamba rather than bidirectional Mamba and did not mention whether input shuffling was applied. Without shuffling during the repeat process, as described in Figure 1, sequential SSM characteristics could lead to information leakage in the noise data generation process. Also, when treating the entire series as an embedding $X^I$, a structure like bidirectional Mamba, which is order-independent, may be more appropriate.

4. What are the basic settings for the forecasting problem in the experiments, such as input lookback length and forecasting horizon? These details are crucial for analyzing the model’s characteristics but are not provided. Additional training settings, like the train-test split, are also missing.

5. Additionally, the term “selective scan mechanism” used in the paper seems somewhat misleading. The “scan” in Mamba (associative scan) refers to hardware parallelization and is unrelated to the algorithm’s selection ability.

---

> ### Author Response · Authors · 2024-11-19
>
> Thank you for your review comments. We have added a substantial number of experiments and included them in the Appendix. We will address your questions one by one based on the experimental results.
>
> Q1: Our method is designed from a perspective similar to that of RNNs. We aim to enhance the selection between the current time step and historical information. If a time step is invalid, more historical information should be retained; if a time step is important, more of the current time step should be retained while forgetting historical information. Matrix A is crucial in this process. This approach is contrary to the model concept of attention structures. We further conducted experiments on attention, proving that it is not suitable for transformer-based models. It may be effective for RNN-based models, but we have not conducted related experiments because our initial goal is to enhance the temporal prediction capability of all Mamba-based models.
>
> Q2: In Appendix C.2, we present the results of comparisons with other pre-training methods. All methods involve representation learning/data augmentation combined with downstream forecasting methods.
>
> Q3: We used vanilla Mamba for pre-training, and the downstream forecasting models include vanilla Mamba and bidirectional Mamba. The sequence order was not changed during repetition, and we believe there is no information leakage. Each time step is independently repeated and noise is added, with gradually increasing noise generated independently. For example, for augmented sequence $[X_{t,0}, X_{t,1}, X_{t,2}, X_{t+1,0}, X_{t+1,1}, X_{t+1,2}]$, the noise on $X_{t,1}$ and $X_{t,2}$ is gradually increasing, and the noise on $X_{t+1,1}$ and $X_{t+1,2}$ is also gradually increasing. However, the noise intensity of $X_{t,1}$ and $X_{t+1,1}$ is the same, and the noise intensity of $X_{t,2}$ and $X_{t+1,2}$ is the same.
>
> Additionally, our method targets a single Mamba-block, using the parameters of one block as the initialization for all mamba-blocks in all Mamba-based models. Therefore, our design follows Mamba's design and is a sequential pre-training process, with the core aim of selecting between the current time step or historical information, similar to RNNs.
>
> Q4: We have supplemented the experimental setup and added comparative experiments. As shown in Appendix C.2, our input is 96, and we compared the experimental results for prediction lengths of $\\{96, 192, 336, 720\\}$.
>
> Q5: "Associative scan" indeed refers to hardware parallelization. We have made changes to the wording in the paper.
>
> W2: In Appendix D, we demonstrate selective enhancement through analysis and visualization. Matrix A determines the memory and forgetting of historical information. Visualization shows that the initialization parameters learned using RCL can capture more complex temporal patterns, allowing for more flexible and complex memory and forgetting of historical information, highlighting key time steps.

---

### Official Review · Reviewer_Fxmp · 2024-11-06

**Soundness:** 1
**Presentation:** 2
**Contribution:** 2
**Rating:** 3
**Confidence:** 4

**Summary:**

The paper develops a pre-training technique for Mamba family of models and applies this to solve time-series forecasting task. Empirical results show lift resulting from the proposed pretraining techniques when applied to Mamba models.

**Strengths:**

- Improved results compared to mamba models without pretraining

**Weaknesses:**

- Experiment tables lack comparison to recent baselines and reporting format is different than the accepted in the literature. It is not clear where the proposed techniques stand against current state of the art.
- Comparison against univariate baselines is missing. Does you approach beat the univariate baselines on these datasets?
- The title claims that the pretraining improves Mamba's selectivity. The word selectivity appears in two places in the paper other than the title: "we freeze the matrix A that controls the mamba’s selectivity" and "Freezing matrix A can retain the enhanced selectivity obtained during pre-training." While A controls mamba’s selectivity and authors hypothesize that pretraining results in a matrix A that improves the selectivity, there is no empirical result in the paper that would help to support the hypothesis. The claim is basically unsupported.
- There are now ablation studies of the two proposed pretraining techniques: Intra-sequence contrast and Repetitive Contrastive Learning.
- Experiments are not clearly described
- The idea of contrastive pretraining does not seem to be entirely novel, even in the time-series literature. The related work section is misleading as it puts a lot of emphasis on contrastive learning in image/textual domains. However, the following works exist in the time-series domain: https://arxiv.org/pdf/2303.11911, https://arxiv.org/pdf/2312.16424, https://arxiv.org/pdf/2206.08496.

**Questions:**

- Can you include the recent baselines from https://arxiv.org/pdf/2405.14616 and https://arxiv.org/pdf/2403.09898 for comparison?
- Can you provide empirical evidence that freezing pretrained matrix A indeed improves Mamba selectivity? Please define selectivity and measure the improvement. Are there qualitative results demonstrating improved selectivity?
- Can you provide ablation of Intra-sequence contrast and Repetitive Contrastive Learning. What happens when only one of them is used? Which one is more powerful? Do they exhibit synergistic effects (more than additive gain)
- Can you provide more details on the experimental setup? For example, how exactly are the metrics in Table 1 computed? Why is the presentation format different than in the other works relying on the same datasets?
- Please extend related work on time series contrastive pretraining works and explain how your work is different and novel.
- Can you include existing time-series pretraining techniques I identified as baselines?
- In related work, the following MLP-based architectures are missing, please include in the discussion and as baselines: https://arxiv.org/pdf/1905.10437, https://arxiv.org/pdf/2201.12886
- The proposed contrastive learning techniques, are they applicable only to Mamba model, or they are more general? If they are general, why are not you exploring their effects on other model families? Is it viable to obtain evidence of accuracy lifts on other model families?

---

> ### Author Response · Authors · 2024-11-19
>
> Thank you for your review comments. We have added a large number of experiments and included them in the Appendix. We will address your questions one by one based on the experimental results.
>
> Q4: First, we would like to explain our work and why the presentation format is different from other works. Our method, RCL, is pre-trained for a Mamba Block to obtain better initialization parameters. We provide these initialization parameters to mamba-based models, using them as the initialization parameters for each mamba block within the mamba-based models. The mamba-based models we chose include mamba, imamba, timemachine, and bi-mamba. To clearly compare performance improvements, we display the results from top to bottom: without pre-trained initialization parameters (w/o), with pre-trained initialization parameters (w), and the percentage improvement after using them (up-rate). Up-rate = (w - wo)/(wo) * 100%. Our work can be summarized as learning a good set of initialization parameters through pre-training, which can enhance the performance of multiple mamba-based models.
>
> Q1: We have added comparisons with recent baselines. The experimental results and analysis are in Appendix C.1. We compared the RCL-enhanced TimeMachine* and Bi-mamba*.
>
> Q2: In Appendix D, we demonstrate selective enhancement through analysis and visualization. Matrix A determines the memory and forgetting of historical information. Visualization shows that the initialization parameters learned using RCL can capture more complex temporal patterns, allowing for more flexible and complex memory and forgetting of historical information, highlighting key time steps.
>
> Q3: We have added two comparative experiments, see Appendix C.3. In ablation experiment 1, we conducted ablations on intra-sequence contrast, inter-sequence contrast, and noise addition. Ablation experiment 2 compared the effects of different noise addition methods on the results. Overall, intra-sequence contrast enhances selection capability, while inter-sequence contrast ensures no deviation from the temporal correlations of the original sequence.
>
> Q5: We have added temporal contrast models in the related work section. 1. Our method is token-level contrast rather than the common sequence-level contrast. 2. Our design perspective is similar to an RNN perspective, which differs from the common attention-based contrast. Generally, the goal of the most of contrastive learning methods is better hidden state, while we aim to enhance the selection of the current time step and historical information. If a time step is invalid, more historical information should be retained; if a time step is important, more of the current time step should be retained while forgetting historical information. Matrix A is crucial in this process. 3. Our method targets the MambaBlock, and a single pre-training result is effective for all Mamba-based models, as verified by experiments. To my knowledge, contrastive learning targeting the mamba structure and parameters is rare.
>
> Q6: In Appendix C.2, we compared with existing temporal pre-training methods and achieved significant advantages. The experimental results also show that other temporal pre-training methods are not particularly effective for Mamba.
>
> Q7: We have added Mlp-based models in Appendix C.1.
>
> Q8: As mentioned earlier, our contrast perspective is similar to an RNN perspective, focusing on identifying whether the current time step is valid and balancing the combination of historical information and the current time step. This is contrary to the model concept of attention structures. We further conducted experiments on attention, proving that it is not suitable for transformer-based models. It may be effective for RNN-based models, but we have not conducted related experiments because our initial goal is to enhance the temporal prediction capability of all Mamba-based models.
>
> Additionally, we have supplemented comparisons of multiple Mamba models across multiple datasets and prediction horizons to demonstrate the effectiveness of our method, RCL. (The experiments on the same dataset use the same set of pre-trained parameters.)

---

### Meta-Review · Area_Chair_sVu2 · 2024-12-21

**Metareview:**

The submission proposes a **Repetitive Contrastive Learning (RCL)** framework to enhance the sequence selection capability of Mamba-based models in time series prediction. The approach involves **Repeating Sequence Augmentation** (adding noise to repeated time steps) and **contrastive pre-training**, employing intra-sequence and inter-sequence contrast to refine Mamba block selectivity. The pre-trained parameters are transferred to Mamba-based models (Mamba, iMamba, TimeMachine, Bi-Mamba), demonstrating performance improvements without significant resource overhead.

## Strengths
1. Token-Level Contrastive Learning: Introduces token-level contrastive learning rather than common sequence-level contrastive learning, improving robustness against noise in long sequences. Reviewer Fxmp acknowledged that token-level contrastive learning, rather than the common sequence-level contrast, is a distinguishing feature of the work.
2. Generalizability Across Models: The proposed pre-trained parameters are applicable to a Mamba-based models, including Mamba, iMamba, TimeMachine, and Bi-Mamba.
3. Computational Efficiency: The method introduces minimal memory overhead and only a linear increase in training time. Reviewer NrLR noted that the single-block pre-training approach avoids significant computational costs, making it resource-efficient.

## Weaknesses
1. Ambiguity of Selectivity Claims: The link between improved selectivity and enhanced performance is not adequately substantiated with theoretical or multi-perspective evidence. Reviewer NrLR questioned whether the improvements result from enhanced selection capability or better latent space representations, suggesting further theoretical analysis is needed.
2. Limited Baseline Comparisons: The paper lacks comparisons with non-Mamba-based baselines and relevant pre-training techniques. Reviewers Fxmp and UGa7 highlighted the need for evaluations against MLP-based models and softmax-attention architectures to strengthen the broader applicability claims.
3. Clarity and Presentation: Several reviewers pointed out issues with clarity, including ambiguous symbols and typographical errors. There exists the inconsistent citation format and undefined symbols detract from the readability.


## Decision
Based on the recommendations from the reviews (Fxmp: 3 (reject), UGa7: 5 (marginally below accept), NrLR: 5 (marginally below accept), 3whs: 6 (marginally above accept)), I recommend rejecting this paper.
While the proposed method has potential and demonstrates empirical improvements on Mamba-based models, there are significant concerns about the clarity of the claims regarding selectivity and the sufficiency of evidence supporting the stated contributions. Specifically:
1. The lack of convincing evidence for enhanced selective ability, which is central to the paper’s contributions.
2. Comparisons with non-Mamba-based baselines and existing time series pretraining methods are inadequate, limiting the broader applicability of the findings.
3. The manuscript’s clarity and presentation fall below ICLR standards, requiring revision to ensure readability and accessibility.

**Additional Comments On Reviewer Discussion:**

During the rebuttal period, reviewers raised concerns, and the authors responded with thorough explanations and extensive additional experiments. Key points included:
1. Baseline Comparisons: Reviewers noted the lack of non-Mamba baselines, such as MLP-based models. The authors added new comparison.
2. Selectivity Claims: Visualizations and analysis were provided, but theoretical or quantitative evidence remained insufficient.
3. Clarity and Presentation: Some improvements were made, but issues with ambiguous symbols and writing persisted.
4. Noise Augmentation: Ablation studies effectively demonstrated the choice of increasing-intensity Gaussian noise.

While the authors improved the paper, unresolved concerns about baselines and selectivity claims influenced the decision.

---

### Decision · Program_Chairs · 2025-01-22

Reject